# Anomaly detection with semi-supervised classification based on risk estimators*

**Le Thi Khanh Hien**  *khanhhiennt@gmail.com*
*Department of Mathematics and Operational Research*
*University of Mons, Belgium*

**Sukanya Patra**  *sukanya.patra@umons.ac.be*
*Department of Computer Science, University of Mons, Belgium*

**Souhaib Ben Taieb**  *souhaib.bentaieb@umons.ac.be*
*Department of Computer Science, University of Mons, Belgium*

**Reviewed on OpenReview:** *https://openreview.net/forum?id=ekvsBtCBUK*

## Abstract

A significant limitation of one-class classification anomaly detection methods is their reliance on the assumption that unlabeled training data only contains normal instances. To overcome this impractical assumption, we propose two novel classification-based anomaly detection methods. Firstly, we introduce a semi-supervised shallow anomaly detection method based on an unbiased risk estimator. Secondly, we present a semi-supervised deep anomaly detection method utilizing a nonnegative (biased) risk estimator. We establish estimation error bounds and excess risk bounds for both risk minimizers. Additionally, we propose techniques to select appropriate regularization parameters that ensure the nonnegativity of the empirical risk in the shallow model under specific loss functions. Our extensive experiments provide evidence of the effectiveness of the risk-based anomaly detection methods.

## 1 Introduction

Anomaly Detection (AD) can be defined as the task of identifying instances that deviates significantly from the majority of the data instances, see e.g., (Chandola et al., 2009; Pang et al., 2020; Ruff et al., 2021) for comprehensive surveys on AD. One important approach for AD is one-class classification Khan & Madden (2014); Tax & Duin (1999). It can be viewed as a specialized binary classification problem aimed at learning a model that distinguishes between positive (normal) and negative (anomalous) classes. This approach assumes that the unlabeled dataset primarily consists of data from the normal class. By utilizing a sufficient amount of normal data, one-class classification AD (OC-AD) methods identify a decision boundary that encompasses all the normal points. For example, the decision boundaries of *shallow* OC-AD methods include a hyperplane with maximum margin Schölkopf et al. (2001), a compact spherical boundary Tax & Duin (1999; 2004), an elliptical boundary (Rousseeuw & Van Driessen, 1999; Rousseeuw, 1985), a pair of subspaces Wang & Cherian (2019), or even a collection of multiple spheres Görnitz et al. (2018). To enhance their applicability in high-dimensional settings, these shallow methods have been extended into *deep* methods Erfani et al. (2016); Ruff et al. (2018).

Unsupervised learning, where only unlabeled data is available, represents the most common setting in AD. Unsupervised AD methods typically assume that the training data consists solely of normal instances Hodge & Austin (2004); Pimentel et al. (2014); Zimek et al. (2012). However, in real-world scenarios, labeled samples may be available alongside the unlabeled dataset, leading to the development of semi-supervised

---

*This work is supported by the FLARACC research project (Federated Learning and Augmented Reality for Advanced Control Centers), funded by the Wallonia region in Belgium.

AD methods, including semi-supervised OC-AD methods Görnitz et al. (2009); Munoz-Mari et al. (2010); Ruff et al. (2020). It is important to note that unsupervised/semi-supervised shallow/deep one-class anomaly detection methods do not explicitly handle mixed unlabeled data. This is because they typically assume that there are no anomalous instances present in the unlabeled dataset, which is impractical in real-world scenarios. Classification methods that handle mixed unlabeled data have been extensively studied in the field of learning with positive and unlabeled examples (LPUE or PU learning). In this context, we have access to information on positive and unlabeled data, but negative data is unavailable. PU learning methods have also been utilized as semi-supervised AD methods Bekker & Davis (2020); Blanchard et al. (2010); Chandola et al. (2009); Ju et al. (2020). It is widely recognized that incorporating labeled anomalies, even if only a few instances, can greatly enhance the AD performance Görnitz et al. (2013); Kiran et al. (2018). Semi-supervised AD methods that consider the availability of negative data have demonstrated highly promising AD performance Han et al. (2022); Ruff et al. (2021; 2020).

To overcome the impractical assumption of OC-AD methods, we adopt the key concept of risk-based PU learning methods du Plessis et al. (2014; 2015); Kiryo et al. (2017); Sakai et al. (2017). These methods propose empirical estimators for the risk associated with the learning problem. In order to improve anomaly detection performance, we focus on the semi-supervised setting where a negative dataset is also available. It is noteworthy that the estimation of risk in anomaly detection is a relatively unexplored subject, distinguished by specific characteristics, especially in terms of error bounds, which are not commonly found in current anomaly detection approaches.

**Contributions**    Our main contributions are summarized as follows.

- Considering AD as a semi-supervised binary classification problem, where we have access to a positive dataset, a negative dataset, and an unlabeled dataset that may contain anomalous examples, we introduce two risk-based AD methods. These methods include a shallow AD approach developed using an unbiased risk estimator and a deep AD method based on a nonnegative risk estimator.

- We develop methods to select suitable regularization that ensures the nonnegativity of the empirical risk in the proposed shallow AD method. This is crucial as negative empirical risk can lead to significant overfitting issues Kiryo et al. (2017).

- We additionally establish estimation error bounds and excess risk bounds for the two risk minimizers, building upon the theoretical findings presented in Kiryo et al. (2017); Niu et al. (2016).

- We conduct extensive experiments on benchmark AD datasets obtained from *Adbench* Han et al. (2022) to compare the performance of our proposed risk-based AD (rAD) methods against various baseline methods.

**Organization**    We discuss related work in Section 2 and provide a brief background on risk estimators in Section 3. We then introduce the two risk estimators that form the basis of our risk-based AD methods in Section 4. Additionally, we present a theoretical analysis in Section 5, present experimental results in Section 6, highlight limitations in Section 7, and conclude the paper in Section 8. All proofs and additional experiments can be found in the supplementary material.

## 2    Related work

We direct readers to Ruff et al. (2021); Roth et al. (2022) for a comprehensive review of recent advancements in anomaly detection techniques (with a particular focus on industrial anomaly detection). We remark that our primary contribution lies in delving deeper into the approach of risk estimation, a technique that remains relatively unexplored within the context of anomaly detection. In the following, we provide a brief overview of the most relevant works related to our proposed risk-based anomaly detection methods.

**AD methods**    Outlier detection (Hawkins, 1980; Hodge & Austin, 2004), novelty detection (Salu, 1988; Pimentel et al., 2014), and AD are closely related topics. In fact, these terms have been used interchangeably, and solutions to outlier detection and novelty detection are often used for AD and vice versa. AD methods can be generally classified into three types (i) density-based methods, which estimate the probability distribution

of normal instances Lecun et al. (2006); Li et al. (2019); Parzen (1962); Pincus (1995), (ii) reconstruction-based methods, which learn a model that fails to reconstruct anomalous instances but succeeds to reconstruct normal instances Dhillon et al. (2004); Hawkins (1974); Hawkins et al. (2002); Huang et al. (2006); Yan et al. (2021), and (iii) one-class classification methods. We refer the readers to Ruff et al. (2021) for a comprehensive review of the three types of AD methods.

**PU learning methods**    Regarding PU learning methods, they can be classified into three categories: biased learning, two-step techniques, and class-prior incorporation. Similarly to one-class classification AD methods, biased PU learning methods make an impractical assumption: they assume/label all unlabeled instances as negative, see e.g., Lee & Liu (2003); Liu et al. (2003). Although the PU learning methods using two-step techniques do not have such assumption, they are heuristics since they first identify "reliable" negative examples and then apply (semi-)supervised learning techniques to the positive labeled instances and the reliable negative instances, see e.g., Li & Liu (2003); Chaudhari & Shevade (2012). To have some theoretical guarantee, the class-prior incorporation methods need to assume that the class priors are known, see e.g., du Plessis et al. (2014); Elkan & Noto (2008); Hsieh et al. (2019). We refer the readers to Bekker & Davis (2020) and the references therein for more details on the three types of PU learning methods. Methods that rely on risk estimators du Plessis et al. (2014; 2015); Kiryo et al. (2017); Sakai et al. (2017) belong to the third category.

## 3    Background on risk estimators

Let $x$ and $y \in \{+1, -1\}$ be random variables with joint density $p(x, y)$. The class-conditional densities are $p_p(x) = P(x|y = +1)$ and $p_n(x) = P(x|y = -1)$. Let $\pi_p = p(y = +1)$ and $\pi_n = p(y = -1)$ be the class-prior probabilities for the positive and negative classes. We have $\pi_p + \pi_n = 1$. Suppose the positive ($\mathcal{P}$), negative ($\mathcal{N}$) and unlabeled ($\mathcal{U}$) data are sampled independently as $(\mathcal{P}) = \{x_i^p\}_{i=1}^{n_p} \sim p_p(x), (\mathcal{N}) = \{x_i^n\}_{i=1}^{n_n} \sim p_n(x), (\mathcal{U}) = \{x_i^u\}_{i=1}^{n_u} \sim p(x)$, where

$$p(x) = \pi_p p_p(x) + \pi_n p_n(x). \tag{1}$$

Given ($\mathcal{P}$), ($\mathcal{N}$) and ($\mathcal{U}$), let us consider a binary classification problem from $x$ to $y$. Suppose $g : \mathbb{R}^d \to \mathbb{R}$ is a decision function that needs to be trained from ($\mathcal{P}$), ($\mathcal{N}$) and ($\mathcal{U}$), and $\ell : \mathbb{R} \times \{+1, -1\} \to \mathbb{R}$ is a loss function that imposes a cost $\ell(t, y)$ if the predicted output is $t$ and the expected output is $y$. Under loss $\ell$, let us denote

$$\mathcal{R}_p^+(g) = \mathbb{E}_{x \sim p_p(x)}[\ell(g(x), +1)], \mathcal{R}_n^+(g) = \mathbb{E}_{x \sim p_n(x)}[\ell(g(x), +1)], \mathcal{R}_u^+(g) = \mathbb{E}_{x \sim p(x)}[\ell(g(x), +1)],$$

$$\mathcal{R}_p^-(g) = \mathbb{E}_{x \sim p_p(x)}[\ell(g(x), -1)], \mathcal{R}_n^-(g) = \mathbb{E}_{x \sim p_n(x)}[\ell(g(x), -1)], \mathcal{R}_u^-(g) = \mathbb{E}_{x \sim p(x)}[\ell(g(x), -1)].$$

Given $\ell$ and assuming that $\pi_p$ is known (in practice, $\pi_p$ can be effectively estimated from ($\mathcal{P}$), ($\mathcal{N}$) and ($\mathcal{U}$) du Plessis & Sugiyama (2013); Saerens et al. (2002)), our goal is to find $g$ that minimizes the risk of $g$, which is defined by

$$\mathcal{R}(g) := \mathbb{E}_{(x,y) \sim p(x,y)}[\ell(g(x), y)] = \pi_p \mathcal{R}_p^+(g) + \pi_n \mathcal{R}_n^-(g). \tag{2}$$

In ordinary classification, the optimal classifier minimizes the expected misclassification rate that corresponds to using zero-one loss in (2), $\ell_{0\text{-}1}(t, y) = 0$ if $ty > 0$ and $\ell_{0\text{-}1}(t, y) = 1$ otherwise. We denote $I(g) = \mathbb{E}_{(x,y) \sim p(x,y)}[\ell_{0\text{-}1}(g(x), y)]$.

**PN risk estimator**    In supervised learning when we have fully labeled data, $\mathcal{R}(g)$ can be approximated by a PN risk estimator $\hat{\mathcal{R}}_{pn}(g) = \pi_p \hat{\mathcal{R}}_p^+(g) + \pi_n \hat{\mathcal{R}}_n^-(g)$, where

$$\hat{\mathcal{R}}_p^+(g) := \frac{1}{n_p} \sum_{i=1}^{n_p} \ell(g(x_i^p), +1), \quad \hat{\mathcal{R}}_n^-(g) := \frac{1}{n_n} \sum_{i=1}^{n_n} \ell(g(x_i^n), -1). \tag{3}$$

**PU risk estimator**    In PU learning when ($\mathcal{N}$) is unavailable, du Plessis et al. (2014; 2015); Kiryo et al. (2017) propose methods to approximate $\mathcal{R}(g)$ from ($\mathcal{P}$) and ($\mathcal{U}$). From (1) we have $\pi_n \mathcal{R}_n^-(g) = \mathcal{R}_u^-(g) - \pi_p \mathcal{R}_p^-(g)$, which implies that

$$\mathcal{R}(g) = \pi_p(\mathcal{R}_p^+ - \mathcal{R}_p^-) + \mathcal{R}_u^-(g). \tag{4}$$

When $\ell$ satisfies the symmetric condition $\ell(t, +1) + \ell(t, -1) = 1$ then we have $\mathcal{R}(g) = 2\pi_p \mathcal{R}_p^+(g) - \pi_p + \mathcal{R}_u^-(g)$, which can be approximated by

$$\hat{\mathcal{R}}_{pu}^{(1)}(g) = 2\pi_p \hat{\mathcal{R}}_p^+(g) - \pi_p + \hat{\mathcal{R}}_u^-(g), \tag{5}$$

where $\hat{\mathcal{R}}_p^+(g)$ is defined in (3) and $\hat{\mathcal{R}}_u^-(g) = \frac{1}{n_u} \sum_{i=1}^{n_u} \ell(g(x_i^u), -1)$, see du Plessis et al. (2014). When $\ell$ satisfies the linear-odd condition $\ell(t, +1) - \ell(t, -1) = -t$ then $\mathcal{R}(g)$ can be approximated by

$$\hat{\mathcal{R}}_{pu}^{(2)}(g) = -\pi_p \frac{1}{n_p} \sum_{i=1}^{n_p} g(x_i^p) + \hat{\mathcal{R}}_u^-(g), \tag{6}$$

see du Plessis et al. (2015). The authors in Kiryo et al. (2017) propose a *non-negative* PU risk estimator

$$\hat{\mathcal{R}}_{pu}^{(3)}(g) = \pi_p \hat{\mathcal{R}}_p^+(g) + \max\{0, \hat{\mathcal{R}}_u^-(g) - \pi_p \hat{\mathcal{R}}_p^-(g)\}, \tag{7}$$

where $\hat{\mathcal{R}}_p^-(g) = \frac{1}{n_p} \sum_{i=1}^{n_p} \ell(g(x_i^p), -1)$. Note that $\hat{\mathcal{R}}_{pu}^{(3)}(g)$ is a biased estimator.

**NU risk estimator**  Similarly, considering NU learning when $(\mathcal{P})$ is unavailable, see Sakai et al. (2017), NU risk estimators can be formulated by combining the equation $\pi_p \mathcal{R}_p^+(g) = \mathcal{R}_u^+(g) - \pi_n \mathcal{R}_n^+(g)$ (which is derived from (1) ) and (2) to obtain

$$\mathcal{R}(g) = -\pi_n(\mathcal{R}_n^+ - \mathcal{R}_n^-) + \mathcal{R}_u^+(g). \tag{8}$$

With a loss satisfying the symmetric condition, we have a nonconvex NU risk estimator

$$\hat{\mathcal{R}}_{nu}^{(1)}(g) = 2\pi_n \hat{\mathcal{R}}_n^-(g) - \pi_n + \hat{\mathcal{R}}_u^+(g), \tag{9}$$

where $\hat{\mathcal{R}}_n^-(g)$ is defined in (3) and $\hat{\mathcal{R}}_u^+(g) = \frac{1}{n_u} \sum_{i=1}^{n_u} \ell(g(x_i^u), +1)$. And with a loss satisfying the linear-odd condition, we get a convex NU risk estimator

$$\hat{\mathcal{R}}_{nu}^{(2)}(g) = \pi_n \frac{1}{n_n} \sum_{i=1}^{n_n} g(x_i^n) + \hat{\mathcal{R}}_u^+(g). \tag{10}$$

Finally, Sakai et al. (2017) proposes to use a linear combination between the PN, the NU, and the PU risk ofdu Plessis et al. (2014; 2015).

## 4 The proposed semi-supervised anomaly detection methods

In the previous section, we presented risk estimators for the PU learning problem where $(\mathcal{N})$ is unavailable. Let us consider the setting where we have access to $(\mathcal{P})$, $(\mathcal{N})$ as well as $(\mathcal{U})$. We perceive semi-supervised AD as a binary classification problem from $x$ to $y \in \{+1, -1\}$, where $+1$ represents the normal class and $-1$ represents the anomalous class. Our goal is to propose risk estimators for the risk in (2). Specifically, we propose two risk estimators for semi-supervised AD that lead to two risk-based AD methods.

If we take a convex combination of (2) and (8), we obtain

$$\begin{aligned} \mathcal{R}(g) &= a(-\pi_n(\mathcal{R}_n^+ - \mathcal{R}_n^-) + \mathcal{R}_u^+(g)) + (1-a)(\pi_p \mathcal{R}_p^+(g) + \pi_n \mathcal{R}_n^-(g)) \\ &= a\mathcal{R}_u^+(g) + (1-a)\pi_p \mathcal{R}_p^+(g) + \pi_n \mathcal{R}_n^-(g) - a\pi_n \mathcal{R}_n^+, \end{aligned} \tag{11}$$

where $a \in (0, 1)$.

The empirical version of (11) yields the following linear combination of PN and NU risk estimators:

$$\hat{\mathcal{R}}_s^{(2)}(g) = a\hat{\mathcal{R}}_u^+(g) + (1-a)\pi_p \hat{\mathcal{R}}_p^+(g) + \pi_n \hat{\mathcal{R}}_n^-(g) - a\pi_n \hat{\mathcal{R}}_n^+(g). \tag{12}$$

While $\hat{\mathcal{R}}_s^{(2)}(g)$ was also considered in Sakai et al. (2017), they only focused on the set of linear classifiers with two specific losses – the (scaled) ramp loss and the truncated (scaled) squared loss (see (Sakai et al., 2017,

Section 4.1)). We consider a more general setting for $\hat{\mathcal{R}}_s^{(2)}$ and also propose methods to choose appropriate regularization for $\hat{\mathcal{R}}_s^{(2)}$ to avoid negative empirical risks. In fact, $\hat{\mathcal{R}}_s^{(2)}$ may take negative values when $\ell$ is unbounded due to the negative term $-a\pi_n\hat{\mathcal{R}}_n^+(g)$. Theorem 1 summarizes the conditions that guarantee a nonnegative objective.

Inspired by $\hat{\mathcal{R}}_{pu}^{(3)}(g)$ in (7), we also propose the following nonnegative risk estimator:

$$\hat{\mathcal{R}}_s^{(1)}(g) = \pi_n\hat{\mathcal{R}}_n^-(g) + (1-a)\pi_p\hat{\mathcal{R}}_p^+(g) + a\max\{0, \hat{\mathcal{R}}_u^+(g) - \pi_n\hat{\mathcal{R}}_n^+(g)\}, \tag{13}$$

where the max term is introduced since $\mathcal{R}_u^+(g) - \pi_n\mathcal{R}_n^+(g) = \pi_p\mathcal{R}_p^+$ must be nonnegative. Note that $\hat{\mathcal{R}}_{pu}^{(3)}(g)$ was designed for the PU learning problem while we propose $\hat{\mathcal{R}}_s^{(1)}(g)$ for the AD problem which often assumes anomalies are rare. In other words, we put more emphasis on $\hat{\mathcal{R}}_u^+(g)$ rather than $\hat{\mathcal{R}}_u^-(g)$.

In Section 5, we will establish the theoretical estimation error bounds and excess risk bounds for the minimizers of both $\min_{g\in\mathcal{G}}\hat{\mathcal{R}}_s^{(1)}(g)$ and $\min_{g\in\mathcal{G}}\hat{\mathcal{R}}_s^{(2)}(g)$, where $\mathcal{G}$ is some class function. We now present the practical optimization problems involved when using $\hat{\mathcal{R}}_s^{(1)}(g)$ and $\hat{\mathcal{R}}_s^{(2)}(g)$.

**Optimization problems**   Suppose $g$ is parameterized by $w$, which needs to be learned from $(\mathcal{P})$, $(\mathcal{N})$ and $(\mathcal{U})$. When $\hat{\mathcal{R}}_s^{(1)}$ in (13) is used, the corresponding optimization problem for AD is

$$\min_w \left\{ \frac{\pi_n}{n_n}\sum_{i=1}^{n_n}\ell(g(x_i^n), -1) + \frac{(1-a)\pi_p}{n_p}\sum_{i=1}^{n_p}\ell(g(x_i^p), +1) \right.$$
$$\left. + a\max\left\{0, \frac{1}{n_u}\sum_{i=1}^{n_u}\ell(g(x_i^u), +1) - \frac{\pi_n}{n_n}\sum_{i=1}^{n_n}\ell(g(x_i^n), +1)\right\} + \lambda\mathbf{R}(w) \right\}, \tag{14}$$

where $\mathbf{R}$ is some regularizer, and $\lambda \geq 0$ is regularization parameter. And when $\hat{\mathcal{R}}_s^{(2)}$ in (12) is used, the corresponding optimization problem is

$$\min_w \left\{ \frac{a}{n_u}\sum_{i=1}^{n_u}\ell(g(x_i^u), +1) + \frac{(1-a)\pi_p}{n_p}\sum_{i=1}^{n_p}\ell(g(x_i^p), +1) \right.$$
$$\left. + \frac{\pi_n}{n_n}\sum_{i=1}^{n_n}\ell(g(x_i^n), -1) - \frac{a\pi_n}{n_n}\sum_{i=1}^{n_n}\ell(g(x_i^n), +1) + \lambda\mathbf{R}(w) \right\}. \tag{15}$$

Unfortunately, the objective of (15) is not guaranteed to be nonnegative due to the negative term $-\frac{a\pi_n}{n_n}\sum_{i=1}^{n_n}\ell(g(x_i^n), +1)$. As pointed out by Kiryo et al. (2017), this can lead to serious overfitting problems. The following theorem provides methods to choose the regularization parameters such that the nonnegativity of the objective of (15) is guaranteed.

**Theorem 1** *Suppose there exist positive constants $b_1$, $b_2$ and $b_3$ such that*

$$\ell(t, -1) - \ell(t, +1) \geq -b_1|t|, \quad and \quad \ell(t, -1) \geq b_2(b_3 - |t|). \tag{16}$$

*(In Table 1 we give examples of loss functions that satisfy (16), see their proofs in the supp. material.)*

*(i) We have*

$$\frac{\pi_n}{n_n}\sum_{i=1}^{n_n}\ell(g(x_i^n), -1) - \frac{a\pi_n}{n_n}\sum_{i=1}^{n_n}\ell(g(x_i^n), +1) \geq (1-a)\pi_n b_2 b_3 - ((1-a)b_2 + ab_1)\frac{\pi_n}{n_n}\sum_{i=1}^{n_n}|g(x_i^n)|.$$

*(ii) If we choose $\lambda$ and $\mathbf{R}$ such that*

$$\lambda\mathbf{R}(w) \geq ((1-a)b_2 + ab_1)\frac{\pi_n}{n_n}\sum_{i=1}^{n_n}|g(x_i^n)| - (1-a)\pi_n b_2 b_3 \tag{17}$$

Table 1: Examples of loss functions satisfying (16)

| Name | $\ell(t,y) = \ell(z)$ with $z = ty$ | Bounded | $(b_1, b_2, b_3)$ |
|---|---|---|---|
| Hinge loss | $\max\{0, 1 - z\}$ | $\times$ | $(2, 1, 1)$ |
| Double hinge loss | $\max\{0, (1 - z)/2, -z\}$ | $\times$ | $(1, 1/2, 1)$ |
| Squared loss | $\frac{1}{2}(z - 1)^2$ | $\times$ | $(2, 1/2, 1/2)$ |
| Modified Huber loss | $\begin{cases} \max\{0, 1 - z\}^2 & \text{if } z \geq -1 \\ -4z & \text{otherwise} \end{cases}$ | $\times$ | $(4, 1, 1/2)$ |
| Logistic loss | $\ln(1 + \exp(-z))$ | $\times$ | $(1, 1, \ln 2)$ |
| Sigmoid loss | $1/(1 + \exp(z))$ | $\checkmark$ | $(1, 1/2, 1)$ |
| Ramp loss | $\max\{0, \min\{1, (1 - z)/2\}\}$ | $\checkmark$ | $(1, 1/2, 1)$ |

*then the objective of* (15) *is always nonnegative.*

*(iii) Consider the specific case $g(x) = \langle w, \phi(x) \rangle$, where $\phi : \mathbb{R}^d \to \mathbb{R}^q$ is a feature map transformation. The following choices of $\lambda$ and $\mathbf{R}$ satisfy* (17).

- $\mathbf{R}(w) = \|w\|_2^2$ *and* $\lambda \geq \frac{((1-a)b_2 + ab_1)^2 \pi_n c^2}{4(1-a)b_2 b_3}$, *where* $c = \max\{\|\phi(x_i^n)\|_2 : i = 1, \ldots, n_n\}$ *(note that, in practice, we can scale the data to have $c = 1$).*

- $\mathbf{R}(w) = \|w\|_1$ *and* $\lambda \geq c_\infty((1-a)b_2 b_3 + ab_1)\pi_n$, *where* $c_\infty = \max\{\|\phi(x_i^n)\|_\infty : i = 1, \ldots, n_n\}$ *(in practice, we can scale the data to have $c_\infty = 1$).*

We consider both a shallow and deep implementation of the rAD method. In the following, $\pi_p^e$ and $\pi_n^e = 1 - \pi_p^e$ will denote estimates of the real class-prior probabilities $\pi_p$ and $\pi_n$, respectively.

**A shallow rAD method**  We plug in $g(x) = \langle w, \phi(x) \rangle$ in (15) (the empirical version of (12)), where $\phi : \mathbb{R}^d \to \mathbb{R}^q$ is a feature map transformation, and choose the regularization method proposed in Theorem 1 (iii). Specifically, we solve the following minimization problem:

$$\min_w \left\{ \frac{a}{n_u} \sum_{i=1}^{n_u} \ell(w^\top \phi(x_i^u), +1) + \frac{(1-a)\pi_p^e}{n_p} \sum_{i=1}^{n_p} \ell(w^\top \phi(x_i^p), +1) \right.$$
$$\left. + \frac{\pi_n^e}{n_n} \sum_{i=1}^{n_n} \ell(w^\top \phi(x_i^n), -1) - \frac{a\pi_n^e}{n_n} \sum_{i=1}^{n_n} \ell(w^\top \phi(x_i^n), +1) + \lambda \mathbf{R}(w) \right\}. \tag{18}$$

**A deep rAD method**  We plug in $g(x) = \phi(x; \mathcal{W})$ in (14) (the empirical version of (13)), where $\mathcal{W}$ is a set of weights of a deep neural network. Specifically, we train a deep neural network by solving the following optimization problem:

$$\min_{\mathcal{W}} \left\{ \frac{\pi_n^e}{n_n} \sum_{i=1}^{n_n} \ell(\phi(x_i^n; \mathcal{W}), -1) + \frac{(1-a)\pi_p^e}{n_p} \sum_{i=1}^{n_p} \ell(\phi(x_i^p; \mathcal{W}), +1) \right.$$
$$\left. + a \max\left\{0, \frac{1}{n_u} \sum_{i=1}^{n_u} \ell(\phi(x_i^u; \mathcal{W}), +1) - \frac{\pi_n^e}{n_n} \sum_{i=1}^{n_n} \ell(\phi(x_i^n; \mathcal{W}), +1)\right\} + \lambda \mathbf{R}(\mathcal{W}) \right\}, \tag{19}$$

where $\mathbf{R}$ can be any regularizer. Note that we focus on these specific implementations but it is also possible to consider a deep model with (15) or a shallow model with (14). It is noteworthy to mention that in our initial numerical findings, we have observed that the shallow model in (18) frequently yields better results compared to the shallow model with (14), while the deep model in (19) outperforms the deep model with (15).

## 5 Risk bounds

In this section, we establish the estimation error bound and the excess risk bound for $\hat{g}^1$ and $\hat{g}^2$ which are the empirical risk minimizers obtained by $\min_{g \in \mathcal{G}} \hat{\mathcal{R}}_s^{(1)}(g)$ and $\min_{g \in \mathcal{G}} \hat{\mathcal{R}}_s^{(2)}(g)$, where $\mathcal{G}$ is a function class.

Let $g^*$ be the true risk minimizer, that is, $g^* = \arg\min_{g \in \mathcal{G}} \mathcal{R}(g)$. Throughout this section, we assume that (i) $\mathcal{G} = \{g \mid \|g\|_\infty \leq C_g\}$ for some constant $C_g$, and (ii) there exists $C_\ell > 0$ such that $\sup_{|t| \leq C_g} \max_y \ell(t, y) \leq C_\ell$. It is worth noting that the set of linear classifiers with bounded norms and feature maps is a special case of Condition (i)

$$\mathcal{G} = \{g(x) = \langle w, \phi(x) \rangle_\mathcal{H} \mid \|w\|_\mathcal{H} \leq C_w, \|\phi(x)\|_\mathcal{H} \leq C_\phi\}, \tag{20}$$

where $\mathcal{H}$ is a Hilbert space, $\phi$ is a feature map, and $C_w$ and $C_\phi$ are positive constants.

Given $g$, $\hat{\mathcal{R}}_s^{(2)}(g)$ is an unbiased estimator of $\mathcal{R}(g)$ but $\hat{\mathcal{R}}_s^{(1)}$ is a biased estimator. The following proposition estimates the bias of $\hat{\mathcal{R}}_s^{(1)}$ (see Inequality (21)) and shows that, for a fixed $g$, $\hat{\mathcal{R}}_s^{(1)}(g)$ and $\hat{\mathcal{R}}_s^{(2)}(g)$ converge to $\mathcal{R}(g)$ with the rate $O\left(\frac{\pi_n}{\sqrt{n_n}} + \frac{\pi_p}{\sqrt{n_p}} + \frac{a}{\sqrt{n_u}}\right)$ (see Inequality (22) and (23)).

**Proposition 1** *Consider a classifier $g$. Suppose there exists $\rho_g > 0$ such that $\mathcal{R}_p^+(g) \geq \rho_g > 0$ and denote $\epsilon_g = a\pi_n C_\ell \exp\left(-\frac{2\pi_p^2 \rho_g^2}{C_\ell^2(1/n_u + \pi_n^2/n_n)}\right)$. Then the bias of $\hat{\mathcal{R}}_s^{(1)}(g)$ satisfies*

$$0 \leq \mathbb{E}[\hat{\mathcal{R}}_s^{(1)}(g)] - \mathcal{R}(g) \leq \epsilon_g. \tag{21}$$

*Moreover, for any $\delta > 0$, we have the following inequalities hold with probability at least $1 - \delta$*

$$|\hat{\mathcal{R}}_s^{(2)}(g) - \mathcal{R}(g)| \leq C_\ell \sqrt{\ln(2/\delta)/2}\left(\frac{(1+a)\pi_n}{\sqrt{n_n}} + \frac{(1-a)\pi_p}{\sqrt{n_p}} + \frac{a}{\sqrt{n_u}}\right), \tag{22}$$

*and*

$$|\hat{\mathcal{R}}_s^{(1)}(g) - \mathcal{R}(g)| \leq C_\ell \sqrt{\ln(2/\delta)/2}\left(\frac{(1+a)\pi_n}{\sqrt{n_n}} + \frac{(1-a)\pi_p}{\sqrt{n_p}} + \frac{a}{\sqrt{n_u}}\right) + \epsilon_g. \tag{23}$$

**Estimation error bound** The Rademacher complexity of $\mathcal{G}$ for a sample of size $n$ drawn from some distribution $q$ (see e.g., Mohri et al. (2018)) is defined by $\mathfrak{R}_{n,q}(\mathcal{G}) := \mathbb{E}_{Z \sim q^n}[\mathbb{E}_\sigma[\sup_{g \in \mathcal{G}}(\frac{1}{n}\sum_{i=1}^n \sigma_i g(Z_i))]]$, where $Z_1, \ldots, Z_n$ are i.i.d random variables following distribution $q$, $Z = (Z_1, \ldots, Z_n)$, $\sigma_1, \ldots, \sigma_n$ are independent random variables uniformly chosen from $\{-1, 1\}$, and $\sigma = (\sigma_1, \ldots, \sigma_n)$. Similarly to (Kiryo et al., 2017, Theorem 4), we can establish the following estimation error bound for $\hat{g}^1$.

**Theorem 2 (Estimation error bound for $\hat{g}^1$)** *We assume that (i) there exists $\rho > 0$ such that $\mathcal{R}_p^+(g) \geq \rho$ for all $g \in \mathcal{G}$, (ii) if $g \in \mathcal{G}$ then $-g \in \mathcal{G}$, and (iii) $t \mapsto \ell(t, 1)$ and $t \mapsto \ell(t, -1)$ are $L_\ell$-Lipschitz continuous over $\{t : |t| \leq C_g\}$. Denote $\epsilon = a\pi_n C_\ell \exp\left(-\frac{2\pi_p^2 \rho^2}{C_\ell^2(1/n_u + \pi_n^2/n_n)}\right)$. For any $\delta > 0$, the following inequality hold with probability at least $1 - \delta$*

$$\mathcal{R}(\hat{g}^1) - \mathcal{R}(g^*) \leq 8(1+a)\pi_n L_\ell \mathfrak{R}_{n_n, p_n}(\mathcal{G}) + 8(1-a)\pi_p L_\ell \mathfrak{R}_{n_p, p_p}(\mathcal{G}) + 8aL_\ell \mathfrak{R}_{n_u, p}(\mathcal{G}) +$$
$$+ 2C_\ell \sqrt{\ln(2/\delta)/2}\left(\frac{(1+a)\pi_n}{\sqrt{n_n}} + \frac{(1-a)\pi_p}{\sqrt{n_p}} + \frac{a}{\sqrt{n_u}}\right) + 2\epsilon. \tag{24}$$

By using basic uniform deviation bound Mohri et al. (2018), the McDiarmid's inequality McDiarmid (1989), and Talagrand's contraction lemma Ledoux & Talagrand (1991), we can prove the following estimation error bound for $\hat{g}^2$.

**Theorem 3 (Estimation error bound for $\hat{g}^2$)** *Assume that $t \mapsto \ell(t, 1)$ and $t \mapsto \ell(t, -1)$ are $L_\ell$-Lipschitz continuous over $\{t : |t| \leq C_g\}$. For any small $\delta > 0$, the following inequality hold with probability at least $1 - \delta$*

$$\mathcal{R}(\hat{g}^2) - \mathcal{R}(g^*) \leq 4(1-a)\pi_p L_\ell \mathfrak{R}_{n_p, p_p}(\mathcal{G}) + 4(a+1)\pi_n L_\ell \mathfrak{R}_{n_n, p_n}(\mathcal{G}) + 4aL_\ell \mathfrak{R}_{n_u, p}(\mathcal{G}) +$$
$$+ 2C_\ell \sqrt{\ln(6/\delta)/2}\left(\frac{(1-a)\pi_p}{\sqrt{n_p}} + \frac{(1+a)\pi_n}{\sqrt{n_n}} + \frac{a}{\sqrt{n_u}}\right). \tag{25}$$

Note that Theorem 3 explicitly states the error bound for $\hat{g}^2$ with any loss function that satisfies the Lipschitz continuity assumption. The (scaled) ramp loss and the truncated (scaled) squared loss considered in Sakai et al. (2017) have $L_\ell = 1/2$.

**Excess risk bound**  The excess risk focuses on the error due to the use of surrogates for the 0-1 loss function. Denote $I^* = \inf_{g \in \mathcal{F}} I(g)$ and $\mathcal{R}^* = \inf_{g \in \mathcal{F}} \mathcal{R}(g)$, where $\mathcal{F}$ is the set of all measurable functions. By using (Bartlett et al., 2006, Theorem 1) (see (42) in the supp. material), Theorem 2, and Theorem 3, we can derive the following excess risk bound for $\hat{g}^1$ and $\hat{g}^2$.

**Corollary 1** *If $\ell$ is a classification-calibrated loss (see Definition 1 in the supp. material), then there exists a convex, invertible, and nondecreasing transformation $\psi_\ell$ with $\psi_\ell(0) = 0$ and the following inequalities hold with probability at least $1 - \delta$*

$$I(\hat{g}^1) - I^* \leq \psi_\ell^{-1}(B_1 + \mathcal{R}(g^*) - \mathcal{R}^*), \quad I(\hat{g}^2) - I^* \leq \psi_\ell^{-1}(B_2 + \mathcal{R}(g^*) - \mathcal{R}^*),$$

*where $B_1$ and $B_2$ are the right hand side of (24) and (25), respectively.*

## 6   Experiments

### A. Experiments with shallow rAD

***Baseline methods and implementation***  We compare rAD with OC-SVM Schölkopf et al. (2001), ECOD Li et al. (2022), COPOD Li et al. (2020), semi-supervised OC-SVM Munoz-Mari et al. (2010), and the PU methods using the risk estimator $\hat{\mathcal{R}}_{pu}(g)$ given in (4). Note that $\hat{\mathcal{R}}_{pu}(g) = \hat{\mathcal{R}}_{pu}^{(1)}(g)$ given in (5) if $\ell$ satisfies the symmetric condition, and $\hat{\mathcal{R}}_{pu}(g) = \hat{\mathcal{R}}_{pu}^{(2)}(g)$ given in (6) if $\ell$ satisfies the linear-odd condition. We implement rAD and PU methods with 3 losses: squared loss, hinge loss, and modified Huber loss. For rAD, we use $l_2$ regularization and take $\phi(x) = x$ in (18), i.e. no kernel is used. We set $a = 0.1$ and $\pi_p^e = 0.8$ ($\pi_n^e = 0.2$) as default values for both the shallow rAD and the PU methods. Note that the real $\pi_n$ of the datasets can be different.

***Datasets***  We test the algorithms on 26 classical anomaly detection benchmark datasets from Han et al. (2022), whose $\pi_n$ ranges from 0.02 to 0.4. The real $\pi_n$ of the datasets are given in the first column of Table 2. We randomly split each dataset 30 times into train and test data with a ratio of 7:3, i.e. we have 30 trials for each dataset. Then, for each trial, we randomly select 5% of the train data to make the labeled data and keep the remaining 95% as unlabeled data.

***Experimental results***  In Table 2, we report the mean and standard error (SE) of the AUC (area under the ROC curve) over 30 trials of the 26 benchmark datasets. We observe that, on average, rAD outperforms the PU methods, OC-SVM methods, ECOD, and COPOD. The difference between the AUC of rAD and that of PU is large on the datasets with $\pi_n \leq 0.2$ but it is small when $\pi_n$ is larger. We also notice that rAD with modified Huber loss often gives better results than rAD with square loss and hinge loss.

***Sensitivity analysis for $\pi_p^e$***  With $a = 0.1$, we run shallow rAD on the 30 trials for $\pi_p^e \in \{1 - \pi_n, 0.9, 0.7, 0.6\}$ ( when $\pi_p^e = 1 - \pi_n$, no approximation is made). The results are reported in Table 3 for 9 benchmark datasets and the results of the 17 remaining datasets are given in Table 5 in the supp. material. From Table 2– 5, we can see that we can obtain good results even if $\pi_p^e$ is different from $\pi_p$. In fact, with $a = 0.1$, we get worse AUC means when $\pi_p^e$ is close to $\pi_p$. The combination $(a, \pi_p^e) = (0.1, 0.8)$ or $(a, \pi_p^e) = (0.1, 0.7)$ seem to be good choices across the datasets. Compared to the other two losses, we found the modified Huber loss to be robust to the values of $\pi_p^e$.

***Sensitivity analysis for $a$***  We run shallow rAD (with fixed $\pi_p^e = 0.8$) on the 30 trials of each dataset for $a \in \{0.3, 0.7, 0.9\}$. The results are reported in Table 4 for the 9 benchmark datasets and the results of the 17 remaining datasets are given in Table 6 in in the supp. material. From Table 2, 4 and 6, we can see that the AUC means do not decrease significantly when we increase $a$ (except for the dataset InternetAds). Hence, shallow rAD with $\pi_p^e = 0.8$ is also robust to different values of $a$.

### B. Experiments with deep rAD

Table 2: Mean (and SE $\times 10^2$) of the AUC over 30 trials. The best means are highlighted in bold. $d$, $n$, and $\pi_n$ denote the feature dimension, the sample size of the dataset, and the ratio of negative samples in the dataset.

| dataset $(d, n, \pi_n)$ | rAD | | | PU | | | OC-SVM | ECOD | COPOD | semi-OC-SVM |
|---|---|---|---|---|---|---|---|---|---|---|
| | square | hinge | m-Huber | square | hinge | m-Huber | | | | |
| pendigits (16, 6870, 0.02) | **0.98**( 0.22) | **0.98**(0.22) | **0.98**( 0.22) | 0.78( 4.79) | 0.78(4.83) | 0.78(4.77) | 0.86(0.31) | 0.92(0.16) | 0.90(0.17) | 0.82(2.48) |
| mammography (6, 11 183, 0.02) | **0.91**( 0.29) | **0.91**(0.29) | **0.91**( 0.29) | 0.87( 1.49) | 0.87(1.49) | 0.87(1.48) | 0.77(0.47) | **0.91**(0.30) | **0.91**(0.29) | 0.61(2.97) |
| optdigits (64, 5216, 0.03) | **1.00**( 0.07) | **1.00**(0.07) | **1.00**( 0.06) | 0.76( 2.93) | 0.75(3.02) | 0.77(2.89) | 0.46(0.53) | 0.60(0.40) | 0.68(0.35) | 0.83(1.82) |
| Stamps (9, 340, 0.09) | 0.90( 1.44) | 0.90(1.24) | 0.90( 1.46) | 0.76( 3.37) | 0.77(3.76) | 0.71(4.21) | 0.65(1.74) | 0.88(0.64) | **0.93**(0.44) | 0.69(3.85) |
| cardio (21, 1831, 0.10) | 0.92( 2.03) | 0.89(2.12) | 0.93( 1.99) | 0.83( 2.09) | 0.81(2.31) | 0.84(1.93) | 0.87(0.32) | **0.94**(0.23) | 0.93(0.21) | 0.79(1.32) |
| InternetAds (1555, 1966, 0.19) | 0.73( 3.00) | **0.87**(0.49) | 0.75( 0.92) | 0.64( 3.45) | 0.77(0.57) | 0.77(0.66) | 0.60(0.54) | 0.68(0.46) | 0.68(0.46) | 0.64(0.97) |
| Cardiotocography (21, 2114, 0.22) | 0.86( 1.32) | 0.84(1.68) | **0.88**( 1.10) | 0.81( 1.86) | 0.79(2.01) | 0.82(1.75) | 0.72(0.41) | 0.78(0.33) | 0.66(0.40) | 0.81(0.80) |
| magic.gamma (10, 19 020, 0.35) | **0.78**( 0.47) | **0.78**(0.49) | **0.78**( 0.45) | **0.78**( 0.69) | 0.77(0.71) | **0.78**(0.68) | 0.56(0.18) | 0.64(0.12) | 0.68(0.11) | 0.54(0.32) |
| SpamBase (57, 4207, 0.40) | **0.94**( 0.15) | **0.94**(0.15) | **0.94**( 0.16) | 0.93( 0.20) | 0.93(0.19) | 0.93(0.21) | 0.54(0.28) | 0.66(0.21) | 0.69(0.21) | 0.64(0.85) |
| satimage-2 (36, 5803, 0.01) | 0.99( 0.17) | 0.99(0.16) | 0.99( 0.17) | 0.80( 4.40) | 0.77(4.29) | 0.82(4.47) | **1.00**(0.09) | 0.96(0.37) | 0.97(0.31) | 0.51(4.52) |
| thyroid (6, 3772, 0.02) | **1.00**( 0.05) | **1.00**(0.04) | **1.00**( 0.05) | 0.86( 2.95) | 0.87(2.95) | 0.86(2.89) | 0.93(0.31) | 0.98(0.07) | 0.94(0.15) | 0.70(2.25) |
| vowels (12, 1456, 0.03) | 0.85( 1.45) | 0.82(1.59) | **0.86**( 1.42) | 0.63( 2.59) | 0.62(2.45) | 0.64(2.66) | 0.72(1.40) | 0.58(1.20) | 0.49(1.12) | 0.69(2.55) |
| Waveform (21, 3443, 0.03) | 0.83( 1.49) | 0.81(1.80) | **0.84**( 1.33) | 0.66( 2.67) | 0.66(2.68) | 0.67(2.64) | 0.67(0.70) | 0.61(0.71) | 0.74(0.53) | 0.78(1.11) |
| CIFAR10-1 (512, 5263, 0.05) | **0.77**( 0.84) | **0.77**(0.84) | **0.77**( 0.86) | 0.59( 1.70) | 0.59(1.83) | 0.59(1.63) | 0.64(0.50) | 0.53(0.45) | 0.49(0.45) | 0.74(0.75) |
| SVHN-1 (512, 10 000, 0.05) | 0.83( 0.46) | 0.83(0.45) | **0.84**( 0.47) | 0.69( 1.42) | 0.69(1.57) | 0.69(1.33) | 0.66(0.27) | 0.65(0.30) | 0.63(0.31) | 0.71(0.77) |
| 20news-1 (768, 2514, 0.05) | 0.64( 1.56) | 0.61(1.14) | **0.68**( 1.70) | 0.51( 1.57) | 0.52(1.21) | 0.53(1.57) | 0.52(0.71) | 0.48(0.76) | 0.48(0.71) | 0.65(1.54) |
| agnews-1 (768, 10000, 0.05) | 0.97( 0.27) | 0.93(0.69) | **0.98**( 0.18) | 0.79( 1.28) | 0.74(1.53) | 0.81(1.12) | 0.76(0.25) | 0.75(0.24) | 0.76(0.24) | 0.89(0.40) |
| amazon (768, 10000, 0.05) | 0.80( 0.76) | 0.76(0.87) | **0.82**( 0.69) | 0.63( 0.98) | 0.60(1.06) | 0.63(0.95) | 0.54(0.40) | 0.51(0.39) | 0.48(0.39) | 0.78(0.56) |
| imdb (768, 10000, 0.05) | 0.82( 0.73) | 0.77(1.00) | **0.83**( 0.65) | 0.63( 1.30) | 0.61(1.22) | 0.65(1.35) | 0.50(0.43) | 0.49(0.42) | 0.50(0.42) | 0.78(0.60) |
| yelp (768, 10000, 0.05) | 0.89( 0.85) | 0.83(1.36) | **0.90**( 0.73) | 0.70( 1.63) | 0.67(1.67) | 0.71(1.55) | 0.61(0.31) | 0.55(0.32) | 0.52(0.33) | 0.82(0.63) |
| mnist (100, 7603, 0.09) | **0.96**( 0.14) | **0.96**(0.15) | **0.96**( 0.14) | 0.92( 0.59) | 0.92(0.57) | 0.92(0.60) | 0.80(0.24) | 0.75(0.23) | 0.78(0.22) | 0.85(0.55) |
| campaign (62, 41 188, 0.11) | **0.85**( 0.16) | **0.85**(0.17) | **0.85**( 0.16) | 0.84( 0.30) | 0.84(0.30) | 0.84(0.30) | 0.68(0.12) | 0.77(0.09) | 0.78(0.09) | 0.77(0.41) |
| vertebral (6, 240, 0.13) | 0.72( 2.57) | **0.75**(2.64) | 0.74( 2.58) | 0.59( 2.60) | 0.58(2.68) | 0.60(2.65) | 0.48(2.18) | 0.43(1.38) | 0.35(1.08) | 0.68(2.59) |
| landsat (36, 6435, 0.21) | 0.73( 0.20) | 0.73(0.21) | 0.73( 0.19) | 0.70( 0.52) | 0.70(0.51) | 0.71(0.51) | 0.35(0.28) | 0.36(0.25) | 0.42(0.24) | **0.76**(0.60) |
| satellite (36, 6435, 0.32) | **0.80**( 0.22) | **0.80**(0.22) | **0.80**( 0.22) | **0.80**( 0.26) | **0.80**(0.25) | **0.80**(0.27) | 0.55(0.30) | 0.59(0.25) | 0.64(0.23) | 0.67(0.72) |
| fault (27, 1941, 0.35) | **0.64**( 0.87) | 0.62(0.76) | **0.64**( 0.91) | 0.58( 1.30) | 0.58(1.27) | 0.59(1.29) | 0.52(0.52) | 0.47(0.47) | 0.46(0.49) | 0.57(0.99) |

Table 3: AUC means of shallow rAD over 30 trials for different $\pi_p^e$. The significant changes in the AUC means are highlighted in bold.

| Dataset | square/$\pi_p^e$ | | | | hinge/$\pi_p^e$ | | | | m-Huber/$\pi_p^e$ | | | |
|---|---|---|---|---|---|---|---|---|---|---|---|---|
| | $1-\pi_n$ | 0.9 | 0.7 | 0.6 | $1-\pi_n$ | 0.9 | 0.7 | 0.6 | $1-\pi_n$ | 0.9 | 0.7 | 0.6 |
| pendigits | 0.96 | 0.98 | 0.98 | 0.98 | 0.94 | 0.98 | 0.98 | 0.98 | 0.97 | 0.98 | 0.98 | 0.98 |
| mammography | 0.90 | 0.91 | 0.91 | 0.91 | 0.90 | 0.91 | 0.91 | 0.91 | 0.90 | 0.91 | 0.91 | 0.91 |
| optdigits | 0.96 | 0.99 | 0.997 | 0.997 | **0.93** | 0.99 | 0.997 | 0.997 | 0.98 | 0.996 | 0.998 | 0.998 |
| Stamps | 0.80 | 0.80 | 0.82 | 0.82 | 0.81 | 0.81 | 0.81 | 0.80 | 0.80 | 0.80 | 0.80 | 0.80 |
| cardio | 0.91 | 0.91 | 0.92 | 0.92 | 0.87 | 0.88 | 0.88 | 0.89 | 0.92 | 0.93 | 0.94 | 0.94 |
| InternetAds | 0.77 | 0.77 | 0.70 | **0.60** | 0.86 | 0.85 | 0.86 | 0.86 | 0.87 | 0.87 | 0.86 | 0.86 |
| Cardiotocography | 0.89 | 0.88 | 0.89 | 0.89 | 0.87 | 0.85 | 0.87 | 0.87 | 0.90 | 0.90 | 0.90 | 0.90 |
| magic.gamma | 0.78 | 0.77 | 0.78 | 0.78 | 0.78 | 0.77 | 0.78 | 0.78 | 0.78 | 0.78 | 0.78 | 0.78 |
| SpamBase | 0.94 | 0.94 | 0.94 | 0.94 | 0.94 | 0.93 | 0.94 | 0.94 | 0.94 | 0.94 | 0.94 | 0.94 |

Table 4: AUC means of shallow rAD over 30 trials for different $a$. The significant changes in the AUC means are highlighted in bold.

| Dataset | square/$a$ | | | hinge/$a$ | | | m-Huber/$a$ | | |
|---|---|---|---|---|---|---|---|---|---|
| | 0.3 | 0.7 | 0.9 | 0.3 | 0.7 | 0.9 | 0.3 | 0.7 | 0.9 |
| pendigits | 0.98 | 0.98 | 0.98 | 0.98 | 0.98 | 0.98 | 0.98 | 0.98 | 0.98 |
| mammography | 0.91 | 0.91 | 0.91 | 0.91 | 0.91 | 0.91 | 0.91 | 0.91 | 0.91 |
| optdigits | 0.997 | 0.995 | 0.99 | 0.996 | 0.995 | 0.99 | 0.997 | 0.996 | 0.99 |
| Stamps | 0.83 | 0.82 | 0.82 | 0.81 | 0.82 | 0.81 | 0.80 | 0.81 | 0.81 |
| cardio | 0.92 | 0.91 | 0.91 | 0.88 | 0.87 | 0.85 | 0.93 | 0.93 | 0.93 |
| InternetAds | 0.79 | **0.69** | **0.62** | 0.87 | 0.85 | **0.77** | 0.83 | **0.71** | **0.65** |
| Cardiotocography | 0.87 | 0.87 | 0.87 | 0.86 | 0.85 | 0.83 | 0.90 | 0.89 | 0.88 |
| magic.gamma | 0.78 | 0.78 | 0.78 | 0.78 | 0.78 | 0.77 | 0.78 | 0.78 | 0.78 |
| SpamBase | 0.94 | 0.94 | 0.93 | 0.94 | 0.94 | 0.93 | 0.94 | 0.94 | 0.94 |

***Baseline methods and implementation*** We compare deep rAD with the Latent Outlier Exposure method (LOE) Qiu et al. (2022), the deep semi-supervised AD method (deep SAD) Ruff et al. (2020) and the PU learning method with nonnegative risk estimator and sigmoid loss (nnPU) Kiryo et al. (2017). For deep SAD and nnPU, we use default hyperparameter settings and network architectures as in their original implementation by the authors. We use the same network architectures as deep SAD for experiments on Fashion-MNIST and MNIST datasets. For experiments on CIFAR-10, the network architecture from nnPU is used. In deep rAD, the optimization problem in (19) is solved using ADAM. We implement 4 losses for deep rAD: squared loss, sigmoid loss, logistic loss, and modified Huber loss. We set $a = 0.1$ and $\pi_p^e = 0.8$ (thus $\pi_n^e = 0.2$) as default values for deep rAD.

***Datasets*** We test the algorithms on 3 benchmark $k$-classes-out datasets: MNIST, Fashion-MNIST, and CIFAR-10 (all have 10 classes). We use AD setups following previous works Chalapathy et al. (2018); Ruff et al. (2020): for each $\pi_n \in \{0.01, 0.05, 0.1, 0.2\}$, we set one of the ten classes to be a positive class, letting the remaining nine classes be anomalies and maintaining the ratio between normal instances and anomaly instances such that the setup has the required $\pi_n$ (so we have 10 setups corresponding to 10 classes). We note that the anomalous data in our generation process can originate from more than one of the nine classes (unlike in the setup of deep SAD where the anomaly is only from one of the nine classes). For each $\pi_n$, we repeat this generation process 2 times to get 20 AD setups (or 20 trials). Then, in each trial, we randomly choose $\gamma_l$ (with $\gamma_l \in \{0.05, 0.1, 0.2\}$) portion of the train data to be labeled and keep the remaining $(1 - \gamma_l)$ portion as unlabeled data. Note that we make the labeled data for nnPU only from normal instances. To make labeled data for deep rAD and deep SAD, $(1 - \pi_n)$ portion is taken from the nnPU labeled data (which contain only normal instances), and the remaining $\pi_n$ portion is taken from the anomalous instances. Hence, the number of labeled anomalous instances for deep rAD and deep SAD is about $(\gamma_l \times \pi_n)$ portion of the train data.

***Experiment results*** In Figure 1, we report the mean and standard deviation (std) of the AUC over 20 trials on the datasets with increasing pollution ratio $\pi_n$ and default $\gamma_l = 0.05$. The results for $\gamma_l \in \{0.1, 0.2\}$ are given in Figures 2 and 3. Figures 1, 2 and 3 show that, on CIFAR-10, LOE performs the best and deep rAD methods on average provide better AUC than deep SAD and nnPU; deep rAD and deep SAD have similar performance when $\pi_n = 0.01$ but their AUC difference is significant when $\pi_n$ is increased. On FMNIST, deep rAD methods, on average, are better than the others when $\pi_n$ is increased but the AUC improvement is small. On MNIST, deep SAD is best; and when either $\pi_n$ or $\gamma_l$ is increased, deep rAD catch up with deep SAD while LOE gives worse AUC than the others. Deep rAD with quadratic loss underperforms the other rAD methods on MNIST and FMNIST. On average, deep rAD with logistic loss performs best among the rAD methods. It is also interesting to note that in the presence of anomalies from multiple classes, the performance of deep SAD degrades over the performance reported in Ruff et al. (2020). The degradation is more severe for CIFAR-10.

To observe the impact of the amount of labeled data, we report the results for the datasets with $\pi_n = 0.1$ and $\gamma_l \in \{0.05, 0.1, 0.2\}$ in Figure 5. We observe that all the semi-supervised methods improve when we increase $\gamma_l$. From $\gamma_l = 0.05$ to $\gamma_l = 0.1$ (i.e., 5% more labeled data), deep rAD methods show a significant improvement in performance.

***Sensitivity analysis for*** $\pi_p^e$ We run deep rAD with $a = 0.1$ on the 20 trials of each dataset for $\pi_p^e \in \{1 - \pi_n, 0.9, 0.7, \pi_n\}$ (when $\pi_p^e = 1 - \pi_n$, it is an exact estimation of $\pi_p$, and when $\pi_p^e = \pi_n$, we can say $\pi_p^e$ is a bad estimation of $\pi_p$). We report the result in Table 7 in the supp. material. Again, we see that $\pi_p^e$ is not necessarily a precise estimation of $\pi_p$; and $(a, \pi_p^e) = (0.1, 0.8)$ or $(a, \pi_p^e) = (0.1, 0.7)$ are good settings. These results are consistent with the results of shallow rAD.

***Sensitivity analysis for*** $a$ We fix $\pi_p^e = 0.8$ and run deep rAD with additional values of $a \in \{0.5, 0.9\}$ ($a = 0.1$ is the default setting). We report the results for the datasets with $\pi_n = 0.1$ and $\gamma_l = 0.05$ in Figure 4. The results for the datasets with other values of $\pi_n$ and $\gamma_l$ are given in the supp. material. We observe that on CIFAR-10, AUC decreases when $a$ is increased; however, the difference is not significant. On FMNIST and MNIST, deep rAD with $\pi_p^e = 0.8$ is quite robust to the change of $a$.

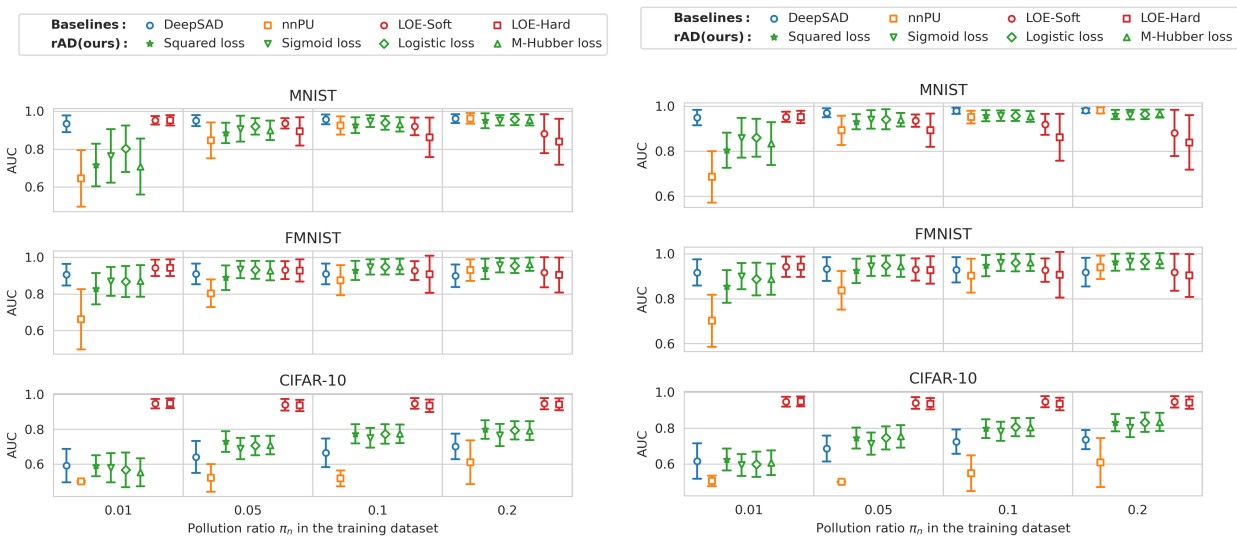

Figure 1: AUC mean and std over 20 trials with various $\pi_n$ and $\gamma_l = 0.05$

Figure 2: AUC mean and SE over 20 trials with various $\pi_n$ and $\gamma_l = 0.1$

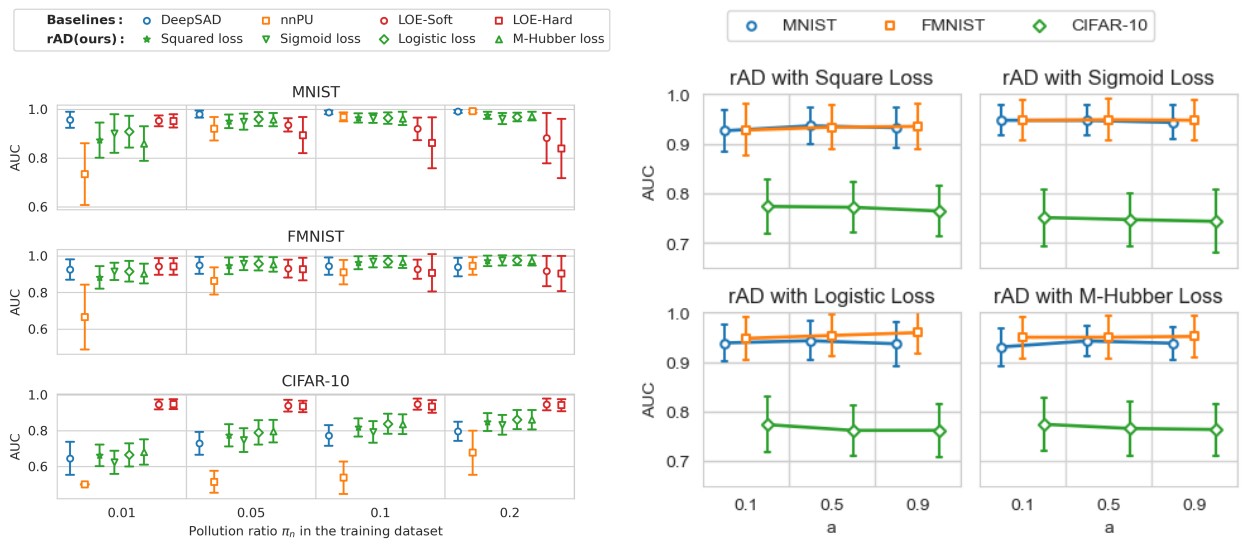

Figure 3: AUC mean and std over 20 trials with various $\pi_n$ and default $\gamma_l = 0.2$

Figure 4: AUC mean and std over 20 trials at various $a$ for the datasets with $\gamma_l = 0.05$ and $\pi_n = 0.1$

## 7 Limitations

On the implementation side, although the experiments have shown that our rAD methods are quite robust to the changes of the parameters $a$ and $\pi_p^e$, we still have to tune them to obtain the best AD performance. Furthermore, solving the optimization problem in (19) is challenging for very large-scale dataset since the max operator does not allow parallel computations. On the theoretical side, although the risk bounds are established for the proposed risk minimizers in Section 5, we still need the assumption that $\pi_p$ and $\pi_n$ are known in advance, which is a limitation.

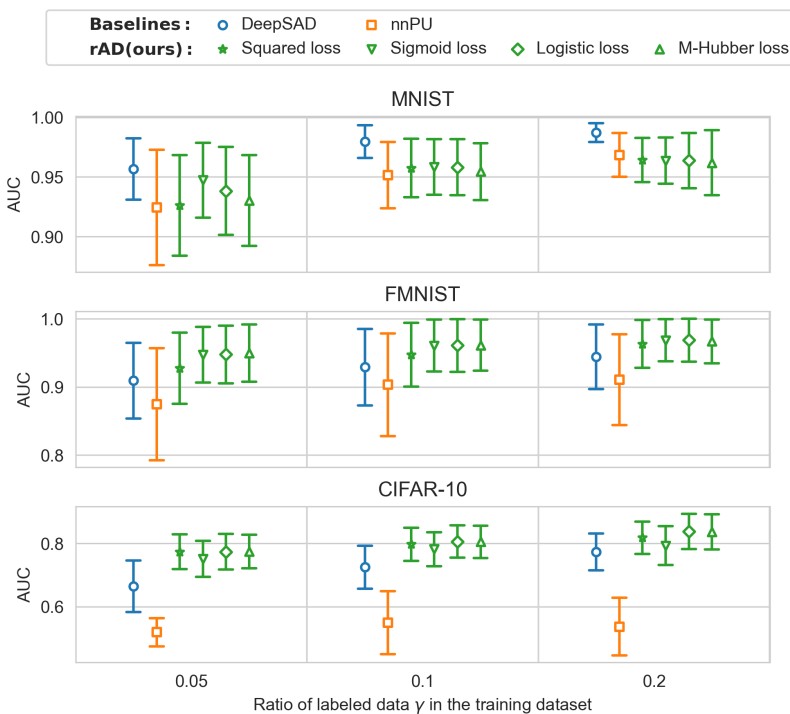

Figure 5: AUC mean and std over 20 trials at various $\gamma_l$ for fixing $\pi_n = 0.1$

## 8 Conclusion

With semi-supervised classification based on risk estimators, we have introduced a shallow AD method equipped with suitable regularization as well as a deep AD method. Theoretically, we have established the estimation error bounds and the excess risk bounds for the two risk minimizers. Empirically, the shallow AD methods show significant improvement over the baseline methods while the deep AD methods compete favorably with the baselines. Let us conclude the paper by giving some possible future research directions that address the limitation given in Section 7. One possible research direction is to develop a method that can learn the best combination of $(a, \pi_p^e)$ from the available data. On the other hand, our experiments have shown that using $a = 0.1$, precise estimation of $\pi_p$ and $\pi_n$ are not necessarily needed to obtain good accuracy in terms of AUC. Hence, another possible research direction would be to study the theoretical bounds of the risk minimizers with $\pi_p$ and $\pi_n$ replaced by some estimates. Finally, investigating effective optimization techniques to tackle the nonconvex Problem (19) is also an important research direction aimed at overcoming the difficulties associated with handling exceedingly large-scale datasets.

**Acknowledgement** We express our sincere appreciation to the reviewers and the action editor for their comments, which greatly helped improve the paper.

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
