## Supplementary material

## A  Technical proofs

### A.1  Proof of Theorem 1

(i) We have

$$\frac{\pi_n}{n_n}\sum_{i=1}^{n_n}\ell(g(x_i^n),-1) - \frac{a\pi_n}{n_n}\sum_{i=1}^{n_n}\ell(g(x_i^n),+1)$$

$$= (1-a)\frac{\pi_n}{n_n}\sum_{i=1}^{n_n}\ell(g(x_i^n),-1) + \frac{a\pi_n}{n_n}\sum_{i=1}^{n_n}(\ell(g(x_i^n),-1) - \ell(g(x_i^n),+1))$$

$$\geq (1-a)\frac{\pi_n}{n_n}\sum_{i=1}^{n_n}b_2(b_3 - |g(x_i^n)|) - a\frac{\pi_n}{n_n}\sum_{i=1}^{n_n}b_1|g(x_i^n)|$$

$$= (1-a)\pi_n b_2 b_3 - ((1-a)b_2 + ab_1)\frac{\pi_n}{n_n}\sum_{i=1}^{n_n}|g(x_i^n)|.$$

(ii) We have (17) is a direct consequence of Theorem 1(i).

(iii) Considering the first case, $\lambda \geq \frac{((1-a)b_2+ab_1)^2\pi_n c^2}{4(1-a)b_2 b_3}$ and $\mathbf{R}(w) = \|w\|_2^2$, we have

$$\lambda\mathbf{R}(w) + (1-a)\pi_n b_2 b_3 \overset{(a)}{\geq} \frac{((1-a)b_2 + ab_1)^2}{4(1-a)b_2 b_3}\frac{\pi_n}{n_n^2}(\sum_{i=1}^{n_n}|g(x_i^n)|)^2 + (1-a)\pi_n b_2 b_3$$

$$\overset{(b)}{\geq} ((1-a)b_2 + ab_1)\frac{\pi_n}{n_n}\sum_{i=1}^{n_n}|g(x_i^n)|,$$

where in (a) we used the property that $|g(x_i^n)| = |\langle w, \phi(x_i^n)\rangle| \leq c\|w\|_2$, and in (b) we used the inequality $u + v \geq 2\sqrt{uv}$ for all nonnegative $u$ and $v$.

Consider the second case, $\mathbf{R}(w) = \|w\|_1$ and $\lambda \geq c_\infty((1-a)b_2 + ab_1)\pi_n$. Note that $|g(x_i^n)| = |\langle w, \phi(x_i^n)\rangle| \leq c_\infty\|w\|_1$. Hence, we have

$$\lambda\mathbf{R}(w) + (1-a)\pi_n b_2 b_3 > c_\infty((1-a)b_2 + ab_1)\pi_n\|w\|_1 \geq ((1-a)b_2 + ab_1)\frac{\pi_n}{n_n}\sum_{i=1}^{n_n}|g(x_i^n)|.$$

**Derivation of $b_1$, $b_2$ and $b_3$ in Table 1**

**Hinge loss.** We have

$$\ell(t,-1) - \ell(t,+1) = \max\{0, 1+t\} - \max\{0, 1-t\} = \begin{cases} t-1 & \text{if } t < -1, \\ 2t & \text{if } -1 \leq t \leq 1, \\ 1+t & \text{if } t > 1, \end{cases}$$

$$\geq -2|t|,$$

and

$$\ell(t,-1) - b_2(1 - |t|) = \max\{0, 1+t\} - b_2(1 - |t|) = \begin{cases} -b_2(1+t) & \text{if } t < -1, \\ t+1 - b_2 - b_2 t & \text{if } -1 \leq t \leq 0, \\ 1+t - b_2 + b_2 t & \text{if } t > 0, \end{cases}$$

Hence, the hinge loss with $b_1 = 2$, $b_2 = 1$ and $b_3 = 1$ satisfies (16).

**Double hinge loss.** Similarly, we have

$$\ell(t,-1) - \ell(t,+1) = \max\{0,(1+t)/2,t\} - \max\{0,(1-t)/2,-t\} = t \geq -|t|,$$

and

$$\ell(t,-1) - b_2(1-|t|) = \max\{0,(1+t)/2,t\} - b_2(1-|t|)$$

$$= \begin{cases} -b_2(1+t) & \text{if } t < -1, \\ t/2 + 1/2 - b_2 - b_2 t & \text{if } -1 \leq t \leq 0, \\ t/2 + 1/2 - b_2 + b_2 t & \text{if } 0 < t \leq 1, \\ t - b_2 + b_2 t & \text{if } t > 1. \end{cases}$$

Hence, the double hinge loss with $b_1 = 1$, $b_2 = 1/2$ and $b_3 = 1$ satisfies (16).

**Square loss.** We have

$$\ell(t,-1) - \ell(t,+1) = \frac{1}{2}(t+1)^2 - \frac{1}{2}(t-1)^2 = 2t \geq -2|t|.$$

Note that when $|t| > 1$ we have $(1/2 - |t|) < 0$, which implies $\ell(t,-1) - 1/2(1/2 - |t|) > 0$. Considering $|t| \leq 1$, when $b_2 = 1/2$ and $b_3 = 1/2$, we have

$$\ell(t,-1) - 1/2(1/2 - |t|) = 1/2(t+1)^2 - 1/2(1/2 - |t|) = \begin{cases} t^2 + 3/2t + 1/4 & \text{if } 0 \leq t \leq 1 \\ t^2 + 1/2t + 1/4 & \text{if } -1 \leq t \leq 0. \end{cases}$$

Hence, the square loss with $b_1 = 2$, $b_2 = 1/2$, $b_3 = 1/2$ satisfies (16).4

**Modified Huber loss.** We have

$$\ell(t,-1) - \ell(t,+1) = \begin{cases} \max\{0,1+t\}^2 & \text{if } t \leq 1 \\ 4t & \text{if } t > 1 \end{cases} - \begin{cases} \max\{0,1-t\}^2 & \text{if } t \geq -1 \\ -4t & \text{if } t < -1 \end{cases}$$

$$= 4t \geq -4|t|.$$

Considering $|t| \leq 1$, when $b_2 = 1$, $b_3 = 1/2$, we have

$$\ell(t,-1) - (1/2 - |t|) = \begin{cases} \max\{0,1+t\}^2 & \text{if } t \leq 1 \\ 4t & \text{if } t > 1 \end{cases} - b_2(1/2 - |t|)$$

$$= \begin{cases} t^2 + 1/2 + t & \text{if } -1 \leq t \leq 0 \\ t^2 + 5/2t + 1/2 & \text{if } 0 < t \leq 1 \end{cases}.$$

Hence the modified Huber loss with $b_1 = 4$, $b_2 = 1$ and $b_3 = 1/2$ satisfies (16).

**Logistic loss.** We have

$$\ell(t,-1) - \ell(t,+1) = t \geq -|t|.$$

When $t \geq 0$ then $\ln(1 + \exp(t)) \geq \ln 2 = b_3 \geq b_3 - |t|$. When $t \leq 0$ we have

$$\ell(t,-1) - b_2(b_3 - |t|) = \ln(1 + \exp(t)) - (\ln 2 + t)$$

$$= \ln\left(\frac{1 + \exp(t)}{2\exp(t)}\right) \geq \ln 1 = 0.$$

Hence the logistic loss with $b_1 = 1$, $b_2 = 1$ and $b_3 = \ln 2$ satisfies (16).

**Sigmoid loss.** When $t > 0$, we have $\ell(t,-1) = \frac{1}{1+\exp(-t)} \geq 1/2 \geq b_2(1-|t|)$. For $t \leq 0$, we have

$$\ell(t,-1) - b_2(1-|t|) = \frac{1}{1+\exp(-t)} - 1/2(1+t) = \frac{1 - 1/2(1+\exp(-t))(1+t)}{1+\exp(-t)}.$$

Note that the function $t \mapsto 1/2(1 + \exp(-t))(1 + t)$ is an increasing function on $(-\infty, 0]$ and its maximum value on $(-\infty, 0]$ is 1. Hence $\ell(t, -1) \geq 1/2(1 - |t|)$. On the other hand, we have

$$\ell(t, -1) - \ell(t, +1) = 2\ell(t, -1) - 1 \geq -|t|.$$

Hence, the sigmoid loss with $b_1 = 1$, $b_2 = 1/2$ and $b_3 = 1$ satisfies (16).

**Ramp loss.** We have

$$\ell(t, -1) - b_2(b_3 - |t|) = \max\{0, \min\{1, (1 + t)/2\}\} - 1/2(1 - |t|)$$

$$= \begin{cases} -1/2(1 + t) & \text{if } t \leq -1 \\ 0 & \text{if } -1 \leq t \leq 0 \\ t & \text{if } 0 < t \leq 1 \\ 1/2 + 1/2t & \text{if } t \geq 1. \end{cases}$$

Hence, $\ell(t, -1) \geq 1/2(1 - |t|)$. On the other hand, we have

$$\ell(t, -1) - \ell(t, +1) = 2\ell(t, -1) - 1 \geq -|t|.$$

Hence, the ramp loss with $b_1 = 1$, $b_2 = 1/2$ and $b_3 = 1$ satisfies (16).

## A.2 Proof of Proposition 1

**Proof of Inequality** (21)

Note that $\mathbb{E}[\hat{\mathcal{R}}_s^{(2)}(g)] = \mathcal{R}(g)$. Considering $\hat{\mathcal{R}}_s^{(1)}(g)$, we have $\hat{\mathcal{R}}_s^{(1)}(g) = \hat{\mathcal{R}}_s^{(2)}(g)$ on

$$\mathcal{M}^+(g) := \{(\mathcal{N}, \mathcal{U}) : \hat{\mathcal{R}}_u^+(g) - \pi_n \hat{\mathcal{R}}_n^+(g) \geq 0\}.$$

Denote $\mathcal{M}^-(g) := \{(\mathcal{N}, \mathcal{U}) : \hat{\mathcal{R}}_u^+(g) - \pi_n \hat{\mathcal{R}}_n^+(g) < 0\}$. We have

$$
\begin{aligned}
\mathbb{E}[\hat{\mathcal{R}}_s^{(1)}(g)] - \mathcal{R}(g) &= \mathbb{E}[\hat{\mathcal{R}}_s^{(1)}(g) - \hat{\mathcal{R}}_s^{(2)}(g)] \\
&= \int_{(\mathcal{N}, \mathcal{U}) \in \mathcal{M}^-(g)} \left(\hat{\mathcal{R}}_s^{(1)}(g) - \hat{\mathcal{R}}_s^{(2)}(g)\right) dF(\mathcal{N}, \mathcal{U}) \\
&= \int_{(\mathcal{N}, \mathcal{U}) \in \mathcal{M}^-(g)} a\left(\pi_n \hat{\mathcal{R}}_n^+(g) - \hat{\mathcal{R}}_u^+(g)\right) dF(\mathcal{N}, \mathcal{U}) \quad (26a) \\
&\leq \sup_{(\mathcal{N}, \mathcal{U}) \in \mathcal{M}^-(g)} a\left(\pi_n \hat{\mathcal{R}}_n^+(g) - \hat{\mathcal{R}}_u^+(g)\right) \int_{(\mathcal{N}, \mathcal{U}) \in \mathcal{M}^-(g)} dF(\mathcal{N}, \mathcal{U}) \\
&= a \sup_{(\mathcal{N}, \mathcal{U}) \in \mathcal{M}^-(g)} \left(\pi_n \hat{\mathcal{R}}_n^+(g) - \hat{\mathcal{R}}_u^+(g)\right) \Pr(\mathcal{M}^-(g)) \\
&\leq a\pi_n C_\ell \Pr(\mathcal{M}^-(g)).
\end{aligned}
\tag{26}
$$

From (26a) we have $\mathbb{E}[\hat{\mathcal{R}}_s^{(1)}(g)] - \mathcal{R}(g) \geq 0$. On the other hand,

$$
\begin{aligned}
\Pr(\mathcal{M}^-(g)) &= \Pr\left(\hat{\mathcal{R}}_u^+(g) - \pi_n \hat{\mathcal{R}}_n^+(g) < 0\right) \\
&\leq \Pr\left(\hat{\mathcal{R}}_u^+(g) - \pi_n \hat{\mathcal{R}}_n^+(g) \leq \pi_p \mathcal{R}_p^+(g) - \pi_p \rho_g\right) \\
&= \Pr\left(\pi_p \mathcal{R}_p^+(g) - (\hat{\mathcal{R}}_u^+(g) - \pi_n \hat{\mathcal{R}}_n^+(g)) \geq \pi_p \rho_g\right) \\
&\leq \exp\left(-\frac{2(\pi_p \rho_g)^2}{n_u(C_\ell/n_u)^2 + n_n(\pi_n C_\ell/n_n)^2}\right) \\
&= \exp\left(-\frac{2\pi_p^2 \rho_g^2}{C_\ell^2(1/n_u + \pi_n^2/n_n)}\right),
\end{aligned}
$$

where we have used McDiarmid's inequality for the last inequality. Therefore, from (26) we have

$$\mathbb{E}[\hat{\mathcal{R}}_s^{(1)}(g)] - \mathcal{R}(g) \leq a\pi_n C_\ell \exp\left(-\frac{2\pi_p^2 \rho_g^2}{C_\ell^2(1/n_u + \pi_n^2/n_n)}\right). \tag{27}$$

**Proof of Inequality** (22) **and** (23)  If an $x_i^n$ is changed then the change of $\hat{\mathcal{R}}_s^{(1)}(g)$ would be no more than $\pi_n(a+1)C_\ell/n_n$. If an $x_i^u$ is changed then the change of $\hat{\mathcal{R}}_s^{(1)}(g)$ would be no more than $aC_\ell/n_u$. And if an $x_i^p$ is changed then the change of $\hat{\mathcal{R}}_s^{(1)}(g)$ would be no more than $(1-a)\pi_p C_\ell/n_p$. For any $\delta > 0$, let

$$\varepsilon = C_\ell \sqrt{\Big(\frac{(1+a)^2\pi_n^2}{n_n} + \frac{(1-a)^2\pi_p^2}{n_p} + \frac{a^2}{n_u}\Big)\ln(2/\delta)/2}.$$

Applying McDiarmid's inequality, we get

$$
\begin{aligned}
&\Pr(|\hat{\mathcal{R}}_s^{(1)}(g) - \mathbb{E}[\hat{\mathcal{R}}_s^{(1)}(g)]| \geq \varepsilon) \\
&\leq 2\exp\left(-\frac{2\varepsilon^2}{n_n(\pi_n(1+a)C_l/n_n)^2 + n_p((1-a)\pi_p C_l/n_p)^2 + n_u(aC_\ell/n_u)^2}\right) \\
&= \delta.
\end{aligned}
\tag{28}
$$

Hence,

$$|\hat{\mathcal{R}}_s^{(1)}(g) - \mathbb{E}[\hat{\mathcal{R}}_s^{(1)}(g)]| \leq \varepsilon \leq C_\ell\sqrt{\ln(2/\delta)/2}\Big(\frac{(1+a)\pi_n}{\sqrt{n_n}} + \frac{(1-a)\pi_p}{\sqrt{n_p}} + \frac{a}{\sqrt{n_u}}\Big)$$

with probability at least $1 - \delta$. Together with (27) and

$$|\hat{\mathcal{R}}_s^{(1)}(g) - \mathcal{R}(g)| \leq |\hat{\mathcal{R}}_s^{(1)}(g) - \mathbb{E}[\hat{\mathcal{R}}_s^{(1)}(g)]| + |\mathbb{E}[\hat{\mathcal{R}}_s^{(1)}(g)] - \mathcal{R}(g)|,$$

we obtain Inequality (23) with probability at least $1 - \delta$.

Similarly, by applying McDiarmid's inequality, we obtain Inequality (22) with probability at least $1 - \delta$.

### A.3  Proof of Theorem 2

Denote $\tilde{\mathcal{R}}_{nu}(g) = \pi_n\hat{\mathcal{R}}_n^-(g) + a\max\{0, \hat{\mathcal{R}}_u^+(g) - \pi_n\hat{\mathcal{R}}_n^+(g)\}$. Note that

$$\hat{\mathcal{R}}_s^{(1)}(g) = (1-a)\pi_p\hat{\mathcal{R}}_p^+ + \tilde{\mathcal{R}}_{nu}(g).$$

We have

$$
\begin{aligned}
\mathcal{R}(\hat{g}^1) - \mathcal{R}(g^*) &= \mathcal{R}(\hat{g}^1) - \hat{\mathcal{R}}_s^{(1)}(\hat{g}^1) + \hat{\mathcal{R}}_s^{(1)}(\hat{g}^1) - \hat{\mathcal{R}}_s^{(1)}(g^*) + \hat{\mathcal{R}}_s^{(1)}(g^*) - \mathcal{R}(g^*) \\
&\overset{(a)}{\leq} |\hat{\mathcal{R}}_s^{(1)}(\hat{g}^1) - \mathcal{R}(\hat{g}^1)| + |\hat{\mathcal{R}}_s^{(1)}(g^*) - \mathcal{R}(g^*)| \\
&\leq 2\sup_{g\in\mathcal{G}}|\hat{\mathcal{R}}_s^{(1)}(g) - \mathcal{R}(g)| \\
&\leq 2\big(\sup_{g\in\mathcal{G}}\big|\hat{\mathcal{R}}_s^{(1)}(g) - \mathbb{E}[\hat{\mathcal{R}}_s^{(1)}(g)]\big| + \sup_{g\in\mathcal{G}}\big|\mathbb{E}[\hat{\mathcal{R}}_s^{(1)}(g)] - \mathcal{R}(g)\big|\big) \\
&\overset{(b)}{\leq} 2\sup_{g\in\mathcal{G}}\big|\hat{\mathcal{R}}_s^{(1)}(g) - \mathbb{E}[\hat{\mathcal{R}}_s^{(1)}(g)]\big| + 2\epsilon \\
&\leq 2(1-a)\pi_p\sup_{g\in\mathcal{G}}\big|\hat{\mathcal{R}}_p^+ - \mathbb{E}[\hat{\mathcal{R}}_p^+]\big| + 2\sup_{g\in\mathcal{G}}\big|\tilde{\mathcal{R}}_{nu}(g) - \mathbb{E}[\tilde{\mathcal{R}}_{nu}(g)]\big| + 2\epsilon,
\end{aligned}
\tag{29}
$$

where we used $\hat{\mathcal{R}}_s^{(1)}(\hat{g}^1) - \hat{\mathcal{R}}_s^{(1)}(g^*) \leq 0$ for (a), and used (21) for (b).

To obtain a bound for $\sup_{g\in\mathcal{G}}\big|\tilde{\mathcal{R}}_{nu}(g) - \mathbb{E}[\tilde{\mathcal{R}}_{nu}(g)]\big|$ we adapt the technique of (Kiryo et al., 2017, Theorem 4). Note that for a fix $g$, $\mathbb{E}[\tilde{\mathcal{R}}_{nu}(g)]$ is a constant. Hence, if an $x_i^n$, or $x_i^u$ is changed then the change of $\sup_{g\in\mathcal{G}}\big|\tilde{\mathcal{R}}_{nu}(g) - \mathbb{E}[\tilde{\mathcal{R}}_{nu}(g)]\big|$ would be the supremum of the change of $\tilde{\mathcal{R}}_{nu}(g)$. By applying McDiarmid's inequality to $\sup_{g\in\mathcal{G}}\big|\tilde{\mathcal{R}}_{nu}(g) - \mathbb{E}[\tilde{\mathcal{R}}_{nu}(g)]\big|$, we have

$$
\begin{aligned}
&\sup_{g\in\mathcal{G}}\big|\tilde{\mathcal{R}}_{nu}(g) - \mathbb{E}[\tilde{\mathcal{R}}_{nu}(g)]\big| - \mathbb{E}\big[\sup_{g\in\mathcal{G}}\big|\tilde{\mathcal{R}}_{nu}(g) - \mathbb{E}[\tilde{\mathcal{R}}_{nu}(g)]\big|\big] \\
&\leq C_\ell\sqrt{\ln(2/\delta)/2}\Big(\frac{(1+a)\pi_n}{\sqrt{n_n}} + \frac{a}{\sqrt{n_u}}\Big)
\end{aligned}
\tag{30}
$$

with probability at least $1 - \delta/2$.

Let $(\mathcal{N}', \mathcal{U}')$ be a ghost sample identical to $(\mathcal{N}, \mathcal{U})$. We have

$$
\begin{aligned}
&\mathbb{E}\left[\sup_{g \in \mathcal{G}} \left|\tilde{\mathcal{R}}_{nu}(g) - \mathbb{E}[\tilde{\mathcal{R}}_{nu}(g)]\right|\right] \\
&= \mathbb{E}_{(\mathcal{N},\mathcal{U})}\left[\sup_{g \in \mathcal{G}} \left|\tilde{\mathcal{R}}_{nu}(g; \mathcal{N}, \mathcal{U}) - \mathbb{E}_{(\mathcal{N}',\mathcal{U}')}[\tilde{\mathcal{R}}_{nu}(g; \mathcal{N}', \mathcal{U}')]\right|\right] \\
&= \mathbb{E}_{(\mathcal{N},\mathcal{U})}\left[\sup_{g \in \mathcal{G}} \left|\mathbb{E}_{(\mathcal{N}',\mathcal{U}')}[\tilde{\mathcal{R}}_{nu}(g; \mathcal{N}, \mathcal{U}) - \tilde{\mathcal{R}}_{nu}(g; \mathcal{N}', \mathcal{U}')]\right|\right] \\
&\leq \mathbb{E}_{(\mathcal{N},\mathcal{U}),(\mathcal{N}',\mathcal{U}')}\left[\sup_{g \in \mathcal{G}} \left|\tilde{\mathcal{R}}_{nu}(g; \mathcal{N}, \mathcal{U}) - \tilde{\mathcal{R}}_{nu}(g; \mathcal{N}', \mathcal{U}')\right|\right],
\end{aligned}
\tag{31}
$$

where we applied Jensen's inequality. Furthermore, we have

$$
\begin{aligned}
&\left|\tilde{\mathcal{R}}_{nu}(g; \mathcal{N}, \mathcal{U}) - \tilde{\mathcal{R}}_{nu}(g; \mathcal{N}', \mathcal{U}')\right| \\
&\leq \pi_n \left|\hat{\mathcal{R}}_n^-(g; \mathcal{N}) - \hat{\mathcal{R}}_n^-(g; \mathcal{N}')\right| \\
&\quad + a\left|\max\{0, \hat{\mathcal{R}}_u^+(g; \mathcal{U}) - \pi_n \hat{\mathcal{R}}_n^+(g; \mathcal{N})\} - \max\{0, \hat{\mathcal{R}}_u^+(g; \mathcal{U}') - \pi_n \hat{\mathcal{R}}_n^+(g; \mathcal{N}')\}\right| \\
&\leq \pi_n \left|\hat{\mathcal{R}}_n^-(g; \mathcal{N}) - \hat{\mathcal{R}}_n^-(g; \mathcal{N}')\right| \\
&\quad + a\left|\hat{\mathcal{R}}_u^+(g; \mathcal{U}) - \hat{\mathcal{R}}_u^+(g; \mathcal{U}')\right| + a\pi_n \left|\hat{\mathcal{R}}_n^+(g; \mathcal{N}) - \hat{\mathcal{R}}_n^+(g; \mathcal{N}')\right|.
\end{aligned}
\tag{32}
$$

Hence, from (31) and (32), we obtain

$$
\begin{aligned}
&\mathbb{E}\left[\sup_{g \in \mathcal{G}} \left|\tilde{\mathcal{R}}_{nu}(g) - \mathbb{E}[\tilde{\mathcal{R}}_{nu}(g)]\right|\right] \\
&\leq \pi_n \, \mathbb{E}_{\mathcal{N},\mathcal{N}'}\left[\sup_{g \in \mathcal{G}} \left|\hat{\mathcal{R}}_n^-(g; \mathcal{N}) - \hat{\mathcal{R}}_n^-(g; \mathcal{N}')\right|\right] + a \, \mathbb{E}_{\mathcal{U},\mathcal{U}'}\left[\sup_{g \in \mathcal{G}} \left|\hat{\mathcal{R}}_u^+(g; \mathcal{U}) - \hat{\mathcal{R}}_u^+(g; \mathcal{U}')\right|\right] \\
&\quad + a\pi_n \, \mathbb{E}_{\mathcal{N},\mathcal{N}'}\left[\sup_{g \in \mathcal{G}} \left|\hat{\mathcal{R}}_n^+(g; \mathcal{N}) - \hat{\mathcal{R}}_n^+(g; \mathcal{N}')\right|\right].
\end{aligned}
\tag{33}
$$

Now by using the same technique of (Kiryo et al., 2017, Lemma 5), we can prove that

$$
\begin{aligned}
\mathbb{E}_{\mathcal{N},\mathcal{N}'}\left[\sup_{g \in \mathcal{G}} \left|\hat{\mathcal{R}}_n^-(g; \mathcal{N}) - \hat{\mathcal{R}}_n^-(g; \mathcal{N}')\right|\right] &\leq 4L_\ell \mathfrak{R}_{n_n, p_n}(\mathcal{G}), \\
\mathbb{E}_{\mathcal{U},\mathcal{U}'}\left[\sup_{g \in \mathcal{G}} \left|\hat{\mathcal{R}}_u^+(g; \mathcal{U}) - \hat{\mathcal{R}}_u^+(g; \mathcal{U}')\right|\right] &\leq 4L_\ell \mathfrak{R}_{n_u, p}(\mathcal{G}), \\
\mathbb{E}_{\mathcal{N},\mathcal{N}'}\left[\sup_{g \in \mathcal{G}} \left|\hat{\mathcal{R}}_n^+(g; \mathcal{N}) - \hat{\mathcal{R}}_n^+(g; \mathcal{N}')\right|\right] &\leq 4L_\ell \mathfrak{R}_{n_n, p_n}(\mathcal{G}).
\end{aligned}
\tag{34}
$$

For completeness, let us provide the proof in the following.

Denote $\tilde{\ell}(t, y) = \ell(t, y) - \ell(0, y)$. Then, $\tilde{\ell}(0, y) = 0$. Note that $t \mapsto \tilde{\ell}(t, y)$ is also $L_\ell$-Lipschitz continuous over $\{t : |t| \leq C_g\}$. Denote $\mathfrak{R}'_{n,q}(\mathcal{G}) := \mathbb{E}_{Z \sim q^n}[\mathbb{E}_\sigma[\sup_{g \in \mathcal{G}} |\frac{1}{n} \sum_{i=1}^n \sigma_i g(Z_i)|]]$. We prove the first inequality of (34), the others can be proved similarly. We have

$$
\begin{aligned}
&\mathbb{E}_{\mathcal{N},\mathcal{N}'}\left[\sup_{g \in \mathcal{G}} \left|\hat{\mathcal{R}}_n^-(g; \mathcal{N}) - \hat{\mathcal{R}}_n^-(g; \mathcal{N}')\right|\right] \\
&= \mathbb{E}_{\mathcal{N},\mathcal{N}'}\left[\sup_{g \in \mathcal{G}} \left|\frac{1}{n_n} \sum_{i=1}^{n_n} \ell(g(x_i^n), -1) - \frac{1}{n_n} \sum_{i=1}^{n_n} \ell(g(x_i'^n), -1)\right|\right] \\
&= \mathbb{E}_{\mathcal{N},\mathcal{N}'}\left[\sup_{g \in \mathcal{G}} \left|\frac{1}{n_n} \sum_{i=1}^{n_n} \left(\tilde{\ell}(g(x_i^n), -1) - \tilde{\ell}(g(x_i'^n), -1)\right)\right|\right] \\
&\overset{(a)}{=} \mathbb{E}_{\mathcal{N},\mathcal{N}',\sigma}\left[\sup_{g \in \mathcal{G}} \left|\frac{1}{n_n} \sum_{i=1}^{n_n} \sigma_i\left(\tilde{\ell}(g(x_i^n), -1) - \tilde{\ell}(g(x_i'^n), -1)\right)\right|\right] \\
&\leq 2\mathfrak{R}'_{n_n, p_n}(\tilde{\ell}(\cdot, -1) \circ \mathcal{G}) \overset{(b)}{\leq} 4L_\ell \mathfrak{R}'_{n_n, p_n}(\mathcal{G}) \overset{(c)}{=} 4L_\ell \mathfrak{R}_{n_n, p_n}(\mathcal{G}),
\end{aligned}
$$

where in (a) we used the property that $\sigma_i$ are independent uniformly distributed random variables taking values in $\{-1, +1\}$, in (b) we use (Ledoux & Talagrand, 1991, Theorem 4.12), and in (c) we use the assumption that both $g$ and $-g$ are in $\mathcal{G}$.

On the other hand, in a similar manner, we can prove that

$$\sup_{g \in \mathcal{G}} \left| \hat{\mathcal{R}}_p^+ - \mathbb{E}[\hat{\mathcal{R}}_p^+] \right| \leq 4L_\ell \mathfrak{R}_{n_p, p_p}(\mathcal{G}) + C_\ell \frac{\sqrt{\ln(2/\delta)/2}}{\sqrt{n_p}} \tag{35}$$

with probability at least $1 - \delta/2$. From (29), (30), (33), (34), and (35), we obtain the result.

### A.4 Proof of Theorem 3

We have

$$\begin{aligned}
\mathcal{R}(\hat{g}^2) - \mathcal{R}(g^*) &= \mathcal{R}(\hat{g}^2) - \hat{\mathcal{R}}_s^{(2)}(\hat{g}^2) + \hat{\mathcal{R}}_s^{(2)}(\hat{g}^2) - \hat{\mathcal{R}}_s^{(2)}(g^*) + \hat{\mathcal{R}}_s^{(2)}(g^*) - \mathcal{R}(g^*) \\
&\overset{(a)}{\leq} \mathcal{R}(\hat{g}^2) - \hat{\mathcal{R}}_s^{(2)}(\hat{g}^2) + \hat{\mathcal{R}}_s^{(2)}(g^*) - \mathcal{R}(g^*) \\
&\leq \sup_{g \in \mathcal{G}} \left( \mathcal{R}(g) - \hat{\mathcal{R}}_s^{(2)}(g) \right) + \sup_{g \in \mathcal{G}} \left( \hat{\mathcal{R}}_s^{(2)}(g) - \mathcal{R}(g) \right) \\
&\overset{(b)}{=} \sup_{g \in \mathcal{G}} \left( \mathbb{E}[\hat{\mathcal{R}}_s^{(2)}(g)] - \hat{\mathcal{R}}_s^{(2)}(g) \right) + \sup_{g \in \mathcal{G}} \left( \hat{\mathcal{R}}_s^{(2)}(g) - \mathbb{E}[\hat{\mathcal{R}}_s^{(2)}(g)] \right)
\end{aligned} \tag{36}$$

where in (a) we have used $\hat{\mathcal{R}}_s^{(2)}(\hat{g}^2) \leq \hat{\mathcal{R}}_s^{(2)}(g^*)$ and in (b) we have used $\mathbb{E}[\hat{\mathcal{R}}_s^{(2)}(g)] = \mathcal{R}(g)$ given $g$.

We have

$$\begin{aligned}
\sup_{g \in \mathcal{G}} \left( \mathbb{E}[\hat{\mathcal{R}}_s^{(2)}(g)] - \hat{\mathcal{R}}_s^{(2)}(g) \right) \leq{}& a \sup_{g \in \mathcal{G}} \left( \mathbb{E}[\hat{\mathcal{R}}_u^+(g)] - \hat{\mathcal{R}}_u^+(g) \right) + (1-a)\pi_p \sup_{g \in \mathcal{G}} \left( \mathbb{E}[\hat{\mathcal{R}}_p^+(g)] - \hat{\mathcal{R}}_p^+(g) \right) \\
&+ \sup_{g \in \mathcal{G}} \left( \mathbb{E}[\pi_n \hat{\mathcal{R}}_n^-(g) - a\pi_n \hat{\mathcal{R}}_n^+] - (\pi_n \hat{\mathcal{R}}_n^-(g) - a\pi_n \hat{\mathcal{R}}_n^+) \right),
\end{aligned} \tag{37}$$

and

$$\begin{aligned}
\sup_{g \in \mathcal{G}} \left( \hat{\mathcal{R}}_s^{(2)}(g) - \mathbb{E}[\hat{\mathcal{R}}_s^{(2)}(g)] \right) \leq{}& a \sup_{g \in \mathcal{G}} \left( \hat{\mathcal{R}}_u^+(g) - \mathbb{E}[\hat{\mathcal{R}}_u^+(g)] \right) + (1-a)\pi_p \sup_{g \in \mathcal{G}} \left( \hat{\mathcal{R}}_p^+(g) - \mathbb{E}[\hat{\mathcal{R}}_p^+(g)] \right) \\
&+ \sup_{g \in \mathcal{G}} \left( (\pi_n \hat{\mathcal{R}}_n^-(g) - a\pi_n \hat{\mathcal{R}}_n^+) - \mathbb{E}[\pi_n \hat{\mathcal{R}}_n^-(g) - a\pi_n \hat{\mathcal{R}}_n^+] \right),
\end{aligned} \tag{38}$$

Applying McDiarmid's inequality to $\sup_{g \in \mathcal{G}} \left( \mathbb{E}[\hat{\mathcal{R}}_u^+(g)] - \hat{\mathcal{R}}_u^+(g) \right)$ we have

$$\sup_{g \in \mathcal{G}} \left( \mathbb{E}[\hat{\mathcal{R}}_u^+(g)] - \hat{\mathcal{R}}_u^+(g) \right) - \mathbb{E}\left[ \sup_{g \in \mathcal{G}} \left( \mathbb{E}[\hat{\mathcal{R}}_u^+(g)] - \hat{\mathcal{R}}_u^+(g) \right) \right] \leq C_\ell \sqrt{\ln(6/\delta)/2} \frac{1}{\sqrt{n_u}} \tag{39}$$

with probability at least $\delta/6$. Moreover, letting $\mathcal{U}'$ be a ghost sample identical to $\mathcal{U}$, we have

$$\begin{aligned}
\mathbb{E}\left[ \sup_{g \in \mathcal{G}} \left( \mathbb{E}[\hat{\mathcal{R}}_u^+(g)] - \hat{\mathcal{R}}_u^+(g) \right) \right] &= \mathbb{E}_{\mathcal{U}}\left[ \sup_{g \in \mathcal{G}} \left( \mathbb{E}_{\mathcal{U}'}[\hat{\mathcal{R}}_u^+(g; \mathcal{U}')] - \hat{\mathcal{R}}_u^+(g; \mathcal{U}) \right) \right] \\
&\overset{(a)}{\leq} \mathbb{E}_{\mathcal{U}, \mathcal{U}'}\left[ \sup_{g \in \mathcal{G}} \left( \hat{\mathcal{R}}_u^+(g; \mathcal{U}') - \hat{\mathcal{R}}_u^+(g; \mathcal{U}) \right) \right] \\
&= \mathbb{E}_{\mathcal{U}, \mathcal{U}'}\left[ \sup_{g \in \mathcal{G}} \left( \frac{1}{n_u} \sum_{i=1}^{n_u} (\ell(g(x'^u_i), +1) - \ell(g(x^u_i), +1)) \right) \right] \\
&\overset{(b)}{\leq} \mathbb{E}_{\mathcal{U}, \mathcal{U}', \sigma}\left[ \sup_{g \in \mathcal{G}} \left( \frac{1}{n_u} \sum_{i=1}^{n_u} \sigma_i (\ell(g(x'^u_i), +1) - \ell(g(x^u_i), +1)) \right) \right] \\
&\leq 2L_\ell \mathfrak{R}_{n_u, p}(\mathcal{G}).
\end{aligned}$$

where we have used the sub-additivity of the supremum in (a) and the property of $\sigma$ in (b). Together with (39) we get

$$\sup_{g \in \mathcal{G}} \left( \mathbb{E}[\hat{\mathcal{R}}_u^+(g)] - \hat{\mathcal{R}}_u^+(g) \right) \leq 2L_\ell \mathfrak{R}_{n_u,p}(\mathcal{G}) + C_\ell \sqrt{\ln(6/\delta)/2} \frac{1}{\sqrt{n_u}} \tag{40}$$

with probability at least $\delta/6$. Similarly, we can prove the following inequalities hold with a probability of at least $1 - \delta/6$

$$
\begin{aligned}
&\sup_{g \in \mathcal{G}} \left( \hat{\mathcal{R}}_u^+(g) - \mathbb{E}[\hat{\mathcal{R}}_u^+(g)] \right) \leq 2L_\ell \mathfrak{R}_{n_u,p}(\mathcal{G}) + C_\ell \sqrt{\ln(6/\delta)/2} \frac{1}{\sqrt{n_u}}, \\
&\sup_{g \in \mathcal{G}} \left( \hat{\mathcal{R}}_p^+(g) - \mathbb{E}[\hat{\mathcal{R}}_p^+(g)] \right) \leq 2L_\ell \mathfrak{R}_{n_p,p_p}(\mathcal{G}) + C_\ell \sqrt{\ln(6/\delta)/2} \frac{1}{\sqrt{n_p}}, \\
&\sup_{g \in \mathcal{G}} \left( \mathbb{E}[\hat{\mathcal{R}}_p^+(g)] - \hat{\mathcal{R}}_p^+(g) \right) \leq 2L_\ell \mathfrak{R}_{n_p,p_p}(\mathcal{G}) + C_\ell \sqrt{\ln(6/\delta)/2} \frac{1}{\sqrt{n_p}}, \\
&\sup_{g \in \mathcal{G}} \left( \pi_n \hat{\mathcal{R}}_n^-(g) - a\pi_n \hat{\mathcal{R}}_n^+ - \mathbb{E}[\pi_n \hat{\mathcal{R}}_n^-(g) - a\pi_n \hat{\mathcal{R}}_n^+] \right) \\
&\qquad \leq 2L_\ell(1+a)\pi_n \mathfrak{R}_{n_n,p_n}(\mathcal{G}) + C_\ell(1+a)\pi_n \sqrt{\ln(6/\delta)/2} \frac{1}{\sqrt{n_n}}, \\
&\sup_{g \in \mathcal{G}} \left( \mathbb{E}[\pi_n \hat{\mathcal{R}}_n^-(g) - a\pi_n \hat{\mathcal{R}}_n^+] - \pi_n \hat{\mathcal{R}}_n^-(g) - a\pi_n \hat{\mathcal{R}}_n^+ \right) \\
&\qquad \leq 2L_\ell(1+a)\pi_n \mathfrak{R}_{n_n,p_n}(\mathcal{G}) + C_\ell(1+a)\pi_n \sqrt{\ln(6/\delta)/2} \frac{1}{\sqrt{n_n}}.
\end{aligned}
\tag{41}
$$

From (36), (37), (38), (40), and (41), we get the result.

# B  Some definitions

**Definition 1** *A loss $\ell$ is said to be classification-calibrated if, for any $\eta \neq \frac{1}{2}$, we have $H_\ell^-(\eta) > H_\ell(\eta)$, where*

$$
\begin{aligned}
H_\ell(\eta) &= \inf_{\alpha \in \mathbb{R}} \left( \eta \ell(\alpha, +1) + (1-\eta)\ell(\alpha, -1) \right), \\
H_\ell^-(\eta) &= \inf_{\alpha \in \mathbb{R}: \alpha(\eta - \frac{1}{2}) \leq 0} \left( \eta \ell(\alpha, +1) + (1-\eta)\ell(\alpha, -1) \right)
\end{aligned}
$$

Examples of classification-calibrated loss include the scaled ramp loss, the hinge loss, and the exponential loss. (Bartlett et al., 2006, Theorem 1) shows that if $\ell$ is a classification-calibrated loss, then there exists a convex, invertible and nondecreasing transformation $\psi_\ell$ with $\psi_\ell(0) = 0$ and $\psi_\ell(I(g) - I^*) \leq \mathcal{R}(g) - \mathcal{R}^*$, which implies that

$$I(g) - I^* \leq \psi_\ell^{-1}(\mathcal{R}(g) - \mathcal{R}^*) = \psi_\ell^{-1}(\mathcal{R}(g) - \mathcal{R}(g^*) + \mathcal{R}(g^*) - \mathcal{R}^*). \tag{42}$$

# C  Additional experiments

## C.1  Additional experiments for shallow rAD

Table 5 reports the mean of the AUC of shallow rAD over the 30 trials for different values of $\pi_p^e$.

Table 6 reports the mean of the AUC of shallow rAD over the 30 trials for different values of $a$.

## C.2  Additional experiments for deep rAD

**Sensitivity analysis for $\pi_p^e$** Table 7 reports the mean and the standard error of the AUC of deep rAD over the 20 trials for different values of $\pi_p^e$.

Table 5: AUC means of shallow rAD over 30 trials for different $\pi_p^e$. The significant changes in the AUC means are highlighted in bold.

| Dataset | square/$\pi_p^e$ | | | | hinge/$\pi_p^e$ | | | | m-Huber/$\pi_p^e$ | | | |
|---|---|---|---|---|---|---|---|---|---|---|---|---|
| | $1-\pi_n$ | 0.9 | 0.7 | 0.6 | $1-\pi_n$ | 0.9 | 0.7 | 0.6 | $1-\pi_n$ | 0.9 | 0.7 | 0.6 |
| thyroid | 0.98 | 0.995 | 0.996 | 0.996 | 0.97 | 0.994 | 0.996 | 0.996 | 0.99 | 0.996 | 0.996 | 0.996 |
| Waveform | **0.74** | 0.82 | 0.84 | 0.84 | **0.70** | **0.78** | 0.83 | 0.83 | **0.77** | 0.84 | 0.85 | 0.85 |
| mnist | 0.96 | 0.96 | 0.97 | 0.97 | 0.96 | 0.96 | 0.96 | 0.96 | 0.97 | 0.97 | 0.97 | 0.97 |
| campaign | 0.85 | 0.85 | 0.85 | 0.85 | 0.85 | 0.85 | 0.85 | 0.85 | 0.85 | 0.85 | 0.85 | 0.85 |
| landsat | 0.74 | 0.74 | 0.74 | 0.74 | 0.74 | 0.73 | 0.74 | 0.74 | 0.74 | 0.74 | 0.74 | 0.74 |
| satellite | 0.80 | 0.80 | 0.80 | 0.80 | 0.80 | 0.80 | 0.80 | 0.80 | 0.81 | 0.80 | 0.80 | 0.80 |
| satimage-2 | 0.97 | 0.98 | 0.98 | 0.98 | **0.93** | 0.98 | 0.98 | 0.98 | 0.98 | 0.99 | 0.99 | 0.99 |
| vowels | **0.77** | 0.85 | 0.87 | 0.87 | **0.69** | **0.77** | 0.85 | 0.85 | 0.85 | 0.88 | 0.88 | 0.88 |
| CIFAR10-1 | **0.69** | 0.73 | 0.77 | 0.77 | **0.66** | **0.71** | 0.76 | 0.76 | **0.71** | 0.74 | 0.77 | 0.77 |
| SVHN-1 | 0.80 | 0.82 | 0.84 | 0.84 | **0.79** | 0.82 | 0.84 | 0.84 | 0.80 | 0.83 | 0.84 | 0.84 |
| 20news-1 | **0.64** | 0.67 | 0.70 | 0.70 | **0.56** | **0.59** | 0.65 | 0.66 | 0.72 | 0.75 | 0.75 | 0.75 |
| agnews-1 | 0.94 | 0.96 | 0.97 | 0.97 | **0.88** | 0.91 | 0.95 | 0.96 | 0.96 | 0.98 | 0.98 | 0.98 |
| amazon | **0.72** | 0.78 | 0.82 | 0.82 | **0.66** | 0.72 | 0.77 | 0.77 | **0.76** | 0.80 | 0.84 | 0.84 |
| imdb | **0.75** | 0.80 | 0.83 | 0.83 | **0.69** | 0.74 | 0.79 | 0.80 | **0.78** | 0.82 | 0.85 | 0.85 |
| yelp | **0.82** | 0.87 | 0.90 | 0.90 | **0.74** | 0.80 | 0.85 | 0.86 | **0.85** | 0.89 | 0.92 | 0.92 |
| vertebral | 0.71 | 0.71 | 0.72 | 0.72 | 0.71 | 0.70 | 0.70 | 0.71 | 0.72 | 0.73 | 0.73 | 0.72 |
| fault | 0.65 | 0.63 | 0.65 | 0.65 | 0.63 | 0.60 | 0.63 | 0.64 | 0.66 | 0.65 | 0.66 | 0.66 |

Table 6: AUC means of shallow rAD over 30 trials for different $a$. The significant changes in the AUC means are highlighted in bold.

| Dataset | square/$a$ | | | hinge/$a$ | | | m-Huber/$a$ | | |
|---|---|---|---|---|---|---|---|---|---|
| | 0.3 | 0.7 | 0.9 | 0.3 | 0.7 | 0.9 | 0.3 | 0.7 | 0.9 |
| thyroid | 0.996 | 0.99 | 0.99 | 0.996 | 0.99 | 0.99 | 0.996 | 0.99 | 0.99 |
| Waveform | 0.84 | 0.81 | 0.80 | 0.82 | 0.80 | **0.77** | 0.85 | 0.83 | 0.81 |
| mnist | 0.97 | 0.96 | 0.96 | 0.96 | 0.96 | 0.96 | 0.97 | 0.96 | 0.96 |
| campaign | 0.85 | 0.85 | 0.85 | 0.85 | 0.85 | 0.85 | 0.85 | 0.85 | 0.85 |
| landsat | 0.74 | 0.74 | 0.73 | 0.74 | 0.74 | 0.73 | 0.74 | 0.74 | 0.74 |
| satellite | 0.80 | 0.80 | 0.80 | 0.80 | 0.80 | 0.80 | 0.80 | 0.81 | 0.81 |
| satimage-2 | 0.98 | 0.99 | 0.98 | 0.98 | 0.99 | 0.98 | 0.99 | 0.99 | 0.98 |
| vowels | 0.87 | 0.85 | 0.83 | 0.85 | 0.81 | 0.78 | 0.88 | 0.87 | 0.86 |
| CIFAR10-1 | 0.76 | 0.73 | 0.70 | 0.75 | 0.74 | 0.72 | 0.76 | 0.72 | **0.69** |
| SVHN-1 | 0.84 | 0.83 | 0.82 | 0.83 | 0.83 | 0.82 | 0.84 | 0.83 | 0.81 |
| 20news-1 | 0.71 | 0.69 | **0.65** | 0.63 | 0.61 | 0.60 | 0.76 | 0.70 | **0.66** |
| agnews-1 | 0.98 | 0.97 | 0.96 | 0.94 | 0.94 | 0.94 | 0.98 | 0.98 | 0.97 |
| amazon | 0.81 | 0.80 | 0.77 | 0.77 | 0.75 | 0.76 | 0.83 | 0.81 | 0.79 |
| imdb | 0.82 | 0.80 | 0.78 | 0.78 | 0.77 | 0.75 | 0.84 | 0.81 | 0.79 |
| yelp | 90 | 0.88 | 0.86 | 0.84 | 0.83 | 0.82 | 0.91 | 0.89 | 0.87 |
| vertebral | 0.72 | 0.73 | 0.73 | 0.73 | 0.74 | 0.73 | 0.74 | 0.75 | 0.74 |
| fault | 0.65 | 0.65 | 0.65 | 0.63 | 0.64 | 0.64 | 0.66 | 0.66 | 0.66 |

Table 7: AUC means (and standard error) of deep rAD over 20 trials for different $\pi_p^e$. The significant changes in the AUC means are highlighted in bold.

| Dataset | Loss | $\pi_p^e = 1 - \pi_n$ | $\pi_p^e = 0.9$ | $\pi_p^e = 0.8$ | $\pi_p^e = 0.7$ | $\pi_p^e = \pi_n$ |
|---|---|---|---|---|---|---|
| MNIST ($\pi_n = 0.01$) | square | **0.66**(0.04) | 0.70(0.03) | 0.72(0.02) | 0.68(0.03) | **0.65**(0.03) |
| | sigmoid | **0.68**(0.03) | 0.76(0.03) | 0.76(0.03) | 0.77(0.03) | 0.77(0.03) |
| | logistic | **0.67**(0.03) | 0.76(0.03) | 0.80(0.03) | 0.77(0.03) | 0.77(0.03) |
| | m-Huber | **0.68**(0.03) | 0.74(0.03) | 0.71(0.03) | 0.72(0.03) | 0.73(0.03) |
| MNIST ($\pi_n = 0.05$) | square | 0.85(0.02) | 0.87(0.01) | 0.89(0.01) | 0.89(0.01) | 0.86(0.01) |
| | sigmoid | 0.88(0.01) | 0.91(0.01) | 0.91(0.01) | 0.93(0.01) | 0.93(0.01) |
| | logistic | 0.87(0.02) | 0.89(0.01) | 0.92(0.01) | 0.92(0.01) | 0.91(0.01) |
| | m-Huber | 0.86(0.01) | 0.88(0.01) | 0.90(0.01) | 0.90(0.01) | 0.87(0.01) |
| MNIST ($\pi_n = 0.1$) | square | 0.92(0.01) | 0.92(0.01) | 0.93(0.01) | 0.93(0.01) | **0.89**(0.01) |
| | sigmoid | 0.94(0.01) | 0.94(0.01) | 0.95(0.01) | 0.95(0.01) | 0.94(0.01) |
| | logistic | 0.93(0.01) | 0.93(0.01) | 0.94(0.01) | 0.94(0.01) | 0.93(0.01) |
| | m-Huber | 0.93(0.01) | 0.93(0.01) | 0.93(0.01) | 0.94(0.01) | 0.92(0.01) |
| MNIST ($\pi_n = 0.2$) | square | 0.95(0.01) | 0.95(0.01) | 0.95(0.01) | 0.96(0.01) | 0.94(0.01) |
| | sigmoid | 0.95(0.01) | 0.95(0.01) | 0.95(0.01) | 0.95(0.01) | 0.95(0.01) |
| | logistic | 0.96(0.01) | 0.96(0.01) | 0.96(0.01) | 0.96(0.01) | 0.96(0.01) |
| | m-Huber | 0.95(0.01) | 0.94(0.01) | 0.95(0.01) | 0.95(0.01) | 0.94(0.01) |
| F-MNIST ($\pi_n = 0.01$) | square | **0.76**(0.02) | 0.80(0.01) | 0.83(0.02) | 0.84(0.02) | **0.78**(0.02) |
| | sigmoid | 0.86(0.02) | 0.87(0.02) | 0.87(0.02) | 0.88(0.02) | 0.88(0.02) |
| | logistic | 0.85(0.02) | 0.87(0.02) | 0.87(0.02) | 0.88(0.02) | 0.87(0.02) |
| | m-Huber | **0.82**(0.02) | 0.88(0.02) | 0.87(0.02) | 0.87(0.02) | **0.83**(0.02) |
| F-MNIST ($\pi_n = 0.05$) | square | **0.84**(0.01) | 0.86(0.01) | 0.89(0.01) | 0.91(0.01) | 0.91(0.01) |
| | sigmoid | 0.93(0.01) | 0.93(0.01) | 0.93(0.01) | 0.94(0.01) | 0.95(0.01) |
| | logistic | 0.91(0.01) | 0.92(0.01) | 0.93(0.01) | 0.93(0.01) | 0.95(0.01) |
| | m-Huber | 0.92(0.01) | 0.94(0.01) | 0.93(0.01) | 0.93(0.01) | 0.93(0.01) |
| F-MNIST ($\pi_n = 0.1$) | square | **0.88**(0.01) | **0.88**(0.01) | 0.93(0.01) | 0.94(0.01) | 0.94(0.01) |
| | sigmoid | 0.94(0.01) | 0.94(0.01) | 0.95(0.01) | 0.95(0.01) | 0.96(0.01) |
| | logistic | 0.94(0.01) | 0.94(0.01) | 0.95(0.01) | 0.95(0.01) | 0.96(0.01) |
| | m-Huber | 0.95(0.01) | 0.95(0.01) | 0.95(0.01) | 0.95(0.01) | 0.95(0.01) |
| F-MNIST ($\pi_n = 0.2$) | square | 0.94(0.01) | 0.92(0.01) | 0.94(0.01) | 0.95(0.01) | 0.96(0.01) |
| | sigmoid | 0.96(0.01) | 0.95(0.01) | 0.96(0.01) | 0.96(0.01) | 0.96(0.01) |
| | logistic | 0.95(0.01) | 0.94(0.01) | 0.95(0.01) | 0.96(0.01) | 0.97(0.01) |
| | m-Huber | 0.96(0.01) | 0.94(0.01) | 0.96(0.01) | 0.96(0.01) | 0.96(0.01) |
| CIFAR-10 ($\pi_n = 0.01$) | square | 0.60(0.01) | 0.60(0.01) | 0.59(0.01) | 0.60(0.01) | 0.59(0.01) |
| | sigmoid | 0.58(0.01) | 0.58(0.02) | 0.58(0.02) | 0.57(0.02) | 0.55(0.02) |
| | logistic | 0.60(0.02) | 0.58(0.02) | 0.57(0.02) | 0.56(0.02) | **0.53**(0.02) |
| | m-Huber | 0.61(0.02) | **0.55**(0.02) | **0.55**(0.02) | **0.55**(0.02) | 0.60(0.02) |
| CIFAR-10 ($\pi_n = 0.05$) | square | 0.73(0.01) | 0.72(0.01) | 0.73(0.01) | 0.72(0.01) | 0.73(0.01) |
| | sigmoid | 0.66(0.02) | 0.68(0.01) | 0.69(0.01) | 0.67(0.01) | 0.69(0.02) |
| | logistic | 0.71(0.01) | 0.71(0.02) | 0.71(0.01) | 0.70(0.01) | 0.69(0.01) |
| | m-Huber | 0.69(0.01) | 0.70(0.01) | 0.71(0.01) | 0.72(0.01) | 0.71(0.01) |
| CIFAR-10 ($\pi_n = 0.1$) | square | 0.77(0.01) | 0.77(0.01) | 0.77(0.01) | 0.78(0.01) | 0.77(0.01) |
| | sigmoid | 0.75(0.01) | 0.75(0.01) | 0.75(0.01) | 0.76(0.01) | 0.73(0.01) |
| | logistic | 0.77(0.01) | 0.77(0.01) | 0.77(0.01) | 0.77(0.01) | 0.76(0.01) |
| | m-Huber | 0.77(0.01) | 0.77(0.01) | 0.77(0.01) | 0.78(0.01) | 0.77(0.01) |
| CIFAR-10 ($\pi_n = 0.2$) | square | 0.80(0.01) | 0.80(0.01) | 0.80(0.01) | 0.80(0.01) | 0.80(0.01) |
| | sigmoid | 0.77(0.01) | 0.74(0.01) | 0.77(0.01) | 0.77(0.01) | 0.77(0.01) |
| | logistic | 0.79(0.01) | 0.78(0.01) | 0.79(0.01) | 0.80(0.01) | 0.79(0.01) |
| | m-Huber | 0.79(0.01) | 0.79(0.01) | 0.79(0.01) | 0.80(0.01) | 0.80(0.01) |

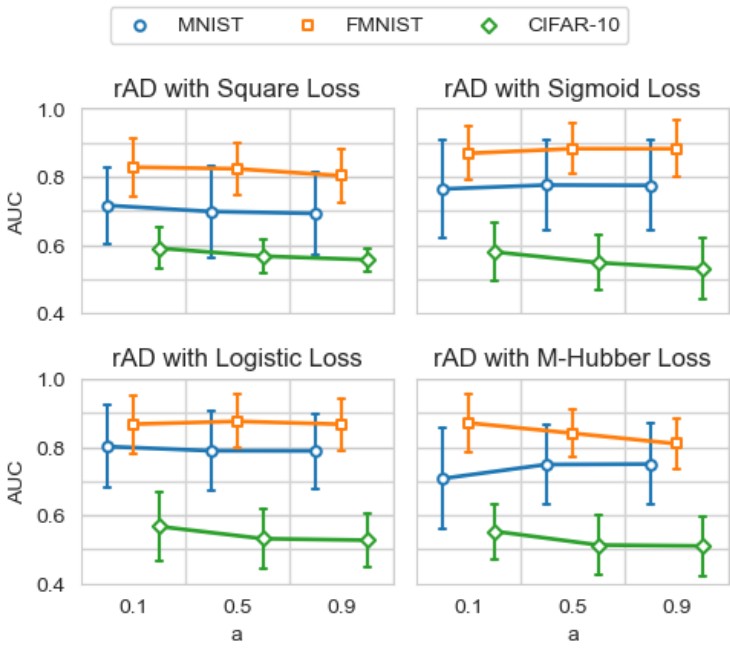

Figure 6: AUC mean and std over 20 trials at various $a$ for the datasets with $\gamma_l = 0.05$ and $\pi_n = 0.01$.

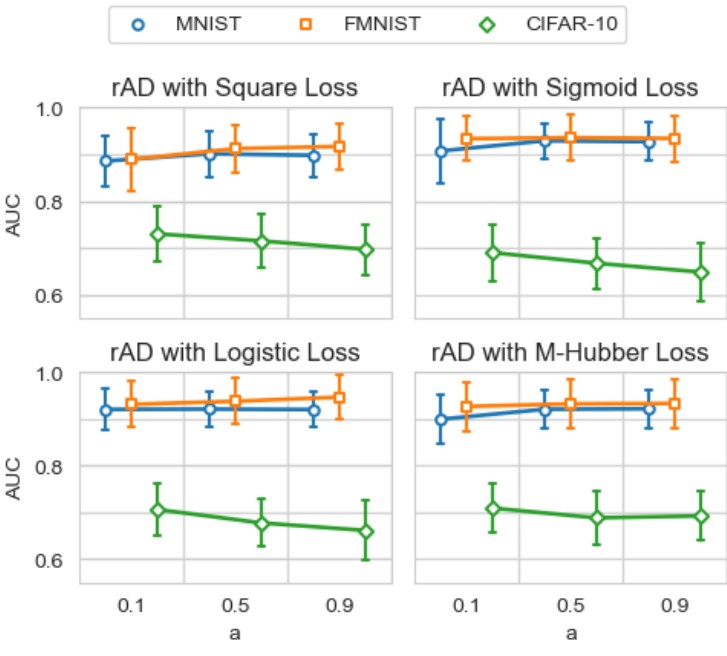

Figure 7: AUC mean and std over 20 trials at various $a$ for the datasets with $\gamma_l = 0.05$ and $\pi_n = 0.05$.

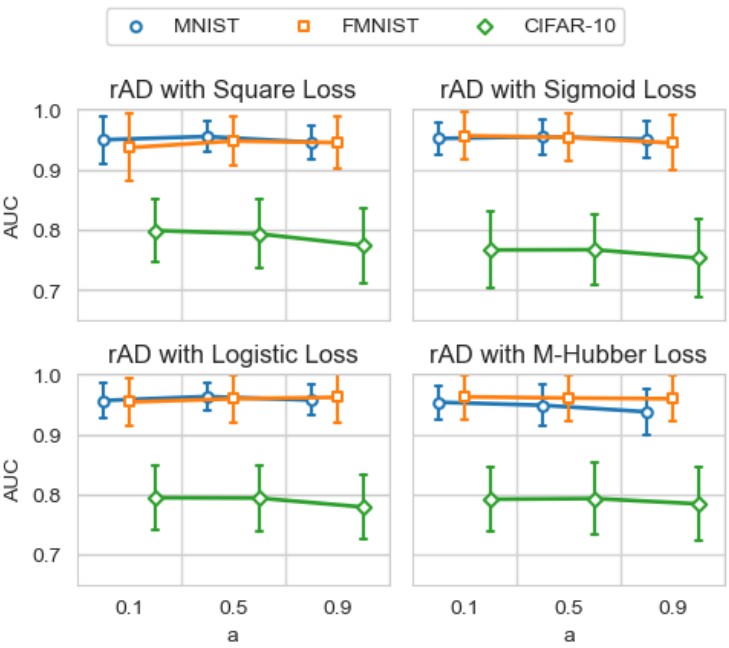

Figure 8: AUC mean and std over 20 trials at various $a$ for the datasets with $\gamma_l = 0.05$ and $\pi_n = 0.2$.

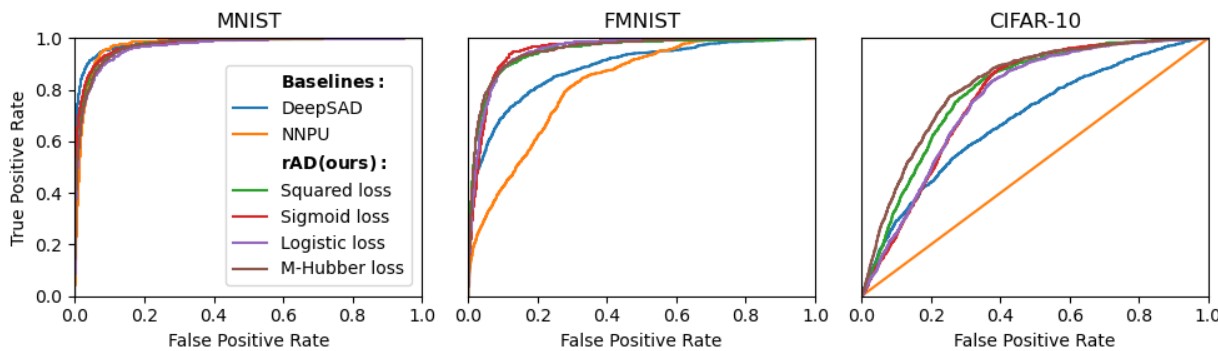

Figure 9: Representative ROC curves for different datasets with $\gamma = 0.05$ and $\pi_n = 0.1$.

**Sensitivity analysis for** $a$    Fixing $\pi_p^e = 0.8$, Figure 6–8 show AUC mean and std of deep rAD with additional values of $a \in \{0.5, 0.9\}$ ($a = 0.1$ is the default setting) on the datasets with $\gamma_l = 0.05$ and $\pi_n = \in \{0.01, 0.05, 0.2\}$

**ROC curves**    Figure 9 shows representative ROC curves obtained by a trial of running the methods (with default settings) on the datasets with $\gamma = 0.05$ and $\pi_n = 0.1$.