# OpenReview forum: "Anomaly detection with semi-supervised classification based on risk estimators"
_TMLR — Accepted by TMLR_

### Review · Reviewer_hBqs · 2023-11-23

**Summary Of Contributions:**

1. The authors approach anomaly detection as a semi-supervised binary classification problem, involving positive, negative, and unlabeled datasets (potentially with anomalies). They introduce two risk-based AD techniques: a shallow AD method utilizing an unbiased risk estimator, and a deep AD strategy using a nonnegative risk estimator.

2. For the shallow AD method, the authors develop techniques to select appropriate regularization, ensuring nonnegative empirical risk.

3. The authors derive estimation error bounds and excess risk bounds for both risk minimizes

4. Through comprehensive experiments on benchmark AD datasets, the authors evaluate the efficacy of our risk-based AD (rAD) methods in comparison to various standard methods.

**Audience:**

Yes

**Claims And Evidence:**

Yes

**Requested Changes:**

I suggest adding more compared methods (baselines) to demonstrate the effectiveness of the proposed method.

**Strengths And Weaknesses:**

Strengths:

1. The paper is easy to follow.

2. The proposed method is novel.

3. The motivation is clear.

Weaknesses:

1. I know the authors discussed several related works in the Introduction. However, it is better to add a subsection to Section 2 to provide more background on the related works.

2.  The baseline methods are very limited. For example, the baselines in Section 6 A are too old. OC-SVM was proposed in 2001. The semi-supervised OC-SVM was proposed in 2010.

3. It seems like most of the experimental results are in supp. materials. I suggest moving them into the main paper.

---

> ### Author Response · Authors · 2024-01-05
> **Response letter**
>
> We sincerely thank the reviewer for taking the time to carefully read the paper and for the feedback that helped us improve our paper. We have revised the paper following the comments of the reviewers. In the following, we provide the detailed response to the reviewer' major comments.
>
> - In the old version, we put the related work in Section 5. Following the reviewer's comment, we improved the related work section and then moved it to Section 2 in the revised version.
>
> - In terms of shallow anomaly detection (AD) methods, despite the introduction of OC-SVM in 2001, it remains among the top unsupervised shallow approaches. Notably, Han et al. (2022) highlighted it as one of the three representative shallow methods alongside Isolation Forest (IForest) (Liu et al., 2008) and Empirical-Cumulative-distribution-based Outlier Detection (ECOD) (Li et al., 2023). According to Table D4 and Section 4.2 of Han et al. (2022), there is no statistically significant superiority among these three methods. Given our paper’s focus on overcoming the impractical assumption of one-class classification AD methods, we opt for OC-
> SVM as a baseline unsupervised method in our experiments. It’s worth noting that, to the best of our knowledge, even though semi-supervised OC-SVM was introduced in 2010, it, along with PU methods, stands as a state-of-the-art choice for semi-supervised shallow AD methods, as discussed in (Ruff et al., 2021).
>
> Concerning deep AD methods, we acknowledge the reviewer’s suggestion to incorporate additional baseline methods in our experiments. However, as our main contribution revolves around a more in-depth exploration of risk estimation, an approach that has been relatively underexplored in the realm of AD, we believe that utilizing deep SAD and the deep PU learning methods as baselines is pertinent for illustrating the effectiveness of our proposed deep risk-based AD method. Furthermore, in the revised version, we also added a discussion about the limitation of the proposed deep rAD model when applying to very large data. It is because solving the optimization problem in (19) is challenging for very large-scale dataset since the max operator does not allow a parallel computation. In the conclusion section of the revised version, we have mentioned that investigating effective optimization techniques to tackle the nonconvex Problem (19) is also an important research direction aimed at overcoming the difficulties associated with handling exceedingly large-scale datasets.
>
> -  Following the reviewer's comment, in the revised version, we have moved some of the experimental results to the main paper.
>
> Reference:
>
> S. Han, X. Hu, H. Huang, M. Jiang, and Y. Zhao. Adbench: Anomaly detection benchmark. In Neural Information Processing Systems (NeurIPS), 2022.
>
> F. T. Liu, K. M. Ting, and Z.-H. Zhou. Isolation forest. In 2008 Eighth IEEE International Conference on Data Mining, pp. 413–422, 2008.
>
> Z. Li, Y. Zhao, X. Hu, N. Botta, C. Ionescu, and G. H. Chen. Ecod: Unsupervised outlier detection using empirical cumulative distribution functions. IEEE Transactions on Knowledge and Data Engineering, 35(12):12181–12193, 2023.
>
> L. Ruff, J. Kauffmann, R. Vandermeulen, G. Montavon, W. Samek, M. Kloft, T. Dietterich, and K.-R.Müller. A unifying review of deep and shallow anomaly detection. Proceedings of the IEEE, PP:1–40, 02 2021.

---

> ### Comment · Action_Editor_kMU9 · 2024-02-19
> **Does the authors' response address your concerns?**
>
> Thank you for reviewing the paper. Could you please check if the authors' response addresses your concerns, especially about baselines?

---

### Review · Reviewer_C1v7 · 2023-12-02

**Summary Of Contributions:**

The paper makes several significant contributions to the field of anomaly detection, particularly in the context of semi-supervised learning. Here is a brief overview of these contributions:

1. **Introduction of Novel Semi-Supervised Anomaly Detection Methods:** The paper presents two new methods for anomaly detection, one based on shallow learning and the other on deep learning. These methods are designed to operate in a semi-supervised framework, addressing the common issue in anomaly detection where it is impractical to assume that all unlabeled training data are normal instances.

2. **Theoretical Foundations:** The authors establish a strong theoretical basis for these methods by deriving estimation error bounds and excess risk bounds. This contribution is significant as it provides a mathematical understanding of the performance and limitations of the proposed methods.

3. **Techniques for Parameter Selection:** The paper proposes techniques for selecting appropriate regularization parameters in the context of these semi-supervised anomaly detection methods. This is a practical contribution that aids in the effective application of these methods.

4. **Extensive Experimental Validation:** The authors conduct a comprehensive set of experiments to demonstrate the effectiveness of the proposed methods. These experiments not only validate the theoretical claims but also show the practical applicability of the methods in various scenarios.

5. **Advancing Semi-Supervised Anomaly Detection:** By addressing the limitations of one-class classification in anomaly detection, the paper contributes to the advancement of semi-supervised learning approaches in this field.

These contributions collectively represent a significant step forward in the anomaly detection domain, particularly in scenarios where labeled data is scarce or imbalanced. The combination of theoretical rigor and practical applicability makes this work a valuable addition to the literature.

**Audience:**

Yes

**Claims And Evidence:**

Yes

**Requested Changes:**

Please discuss more the large data and application scope.

**Strengths And Weaknesses:**

### Strengths

1. **New Methods:** The paper introduces two new approaches for finding anomalies using semi-supervised learning.

2. **Solid Theoretical Background:** The authors provide a detailed mathematical explanation for their methods. This includes calculations that show how well the methods should work and their limitations.

3. **Helpful Techniques for Choosing Parameters:** The paper offers practical ways to choose certain settings (regularization parameters) for these methods, making them easier to use effectively.

### Weaknesses

1. **Handling Large Data:** The paper could talk more about how these methods work with very big sets of data.

2. **Limited Application Scope:** Showing how these methods can be used in specific real-life situations would help understand their practical use.

---

> ### Author Response · Authors · 2024-01-05
> **Response letter**
>
> We sincerely thank the reviewer for taking the time to carefully read the paper and for the feedback that helped us improve our paper. We have revised the paper following the comments of the reviewers. In the following, we provide the detailed response to the reviewer' major comments.
>
> - Following the reviewer's comment, in Section 7 of the revised version, we added a discussion about the limitation of the proposed deep rAD model when applying to very large data. Specifically, we note that solving the optimization problem in (19) is challenging for very large-scale dataset since the max operator does not allow parallel computation. Furthermore, in the conclusion section of the revised version, we have mentioned that investigating effective optimization techniques to tackle the nonconvex Problem (19) is an important research direction aimed at overcoming the difficulties associated with handling exceedingly large-scale datasets.
>
> - In terms of real-life application scope, we have cited important surveys Chandola et al., 2009; Pang et al., 2020; Ruff et al., 2021; Han et al., 2022, where readers can find numerous applications of AD methods across a variety of domains. In the experiments of our proposed shallow rAD method, we use 26 benchmark datasets from Han et al., 2022, which are real-life datasets in various domains such as Healthcare, Finance, Astronautics, etc., see Table B1 of Han et al., 2022. We admit that the deep rAD method needs more investigation to be applicable to real-life very large-scale datasets. This is one of the research directions that we mention in the revised version.
>
> Reference:
>
> V. Chandola, A. Banerjee, and V. Kumar. Anomaly detection. ACM Computing Surveys, 41(3):1–58, 7 2009.
>
> G. Pang, C. Shen, L. Cao, and A. V. D. Hengel. Deep Learning for Anomaly Detection: A Review. ACM Computing Surveys, 54(2), 7 2020.
>
> L. Ruff, J. Kauffmann, R. Vandermeulen, G. Montavon, W. Samek, M. Kloft, T. Dietterich, and K.-R.Müller. A unifying review of deep and shallow anomaly detection. Proceedings of the IEEE, PP:1–40, 02 2021.
>
> S. Han, X. Hu, H. Huang, M. Jiang, and Y. Zhao. Adbench: Anomaly detection benchmark. In Neural Information Processing Systems (NeurIPS), 2022.

---

### Review · Reviewer_5ape · 2024-01-01

**Summary Of Contributions:**

This paper focuses on the semi-supervised anomaly detection. Different from prior works, this paper assumes that a positive set, a negative set and an unlabeled set are attained during training. Directly derived from the risk estimators, the authors propose two types of methods: a shallow one with unbiased risk estimator and a deep model with biased risk estimator. Complete theoretical analysis and extensive experiments are provided to verify the model effectiveness.

**Audience:**

Yes

**Broader Impact Concerns:**

This is an theoritical paper, which can insipre some works in anomaly detection area.

**Claims And Evidence:**

Yes

**Requested Changes:**

The authors need to clarify the experiment setting, necessity of theoretical analysis and the relationship between two types of methods. The writing in Section 2 can also be improved.

**Strengths And Weaknesses:**

### Strengths

-	This paper is well-written.
-	Extensive experiments and detailed theoretical analysis are included.
-	The proposed method is reasonable.

### Weaknesses

1.	Experiment setting.

Different from the previous methods, this paper also assumes that we can access the negative set, which is usually hard to label. They should present the number of known negative samples. This information is helpful for us to estimate the labeling overload.

2.	Necessity of the theoretical analysis.

In theorem 1, the authors prove that the risk defined in Eq (15) is non-negative. I appreciate the proof and theoretical analysis. However, since they use Eq (15) as a loss function, does it have to be nonnegative? The necessity of Theorem 1 should be clarified.
In my opinion, this prove ensures that we can justify how well is the model trained based on the training loss, which requires the training loss curve.

3.	Two types of methods.

As the authors state, this paper presents two types of methods. It is important to analyze more about their differences and relationship.

-	Why do the authors only apply the Eq (15) to the shallow method and the Eq (14) to the deep one? Although the authors state that the cross application is also doable (last sentence in Section 3), it is still necessary to check the final performance.

-	Is the Eq (15) inherently better than Eq (14)? If we come across the anomaly detection task, how to choose the method in practice?

4.	Notations in Section 2.
The upper notation for risk is just pure digits, making it hard to read. A more accessible notation is expected.

After rebuttal, I have changed my score.

---

> ### Author Response · Authors · 2024-01-05
> **Response letter**
>
> We sincerely thank the reviewer for taking the time to carefully read the paper and for the feedback that helped us improve our paper. We have revised the paper following the comments of the reviewers. In the following, we provide the detailed response to the reviewer' major comments.
>
> - We indicated in Table 2 and Table 5 the value of $\pi_n$, which denotes the real ratio of negative samples in the 26 benchmark real datasets in the experiments for shallow rAD method. In the experiments for deep rAD method, we described in the paragraph "Datasets" how to set up the datasets with $\pi_n\in \{0.01, 0.05, 0.1, 0.2\}$.
>
> - We note that the risk $\mathcal R (g)$ defined in (2) ($\mathcal R (g)$ is rewritten in an equivalent form in (11)) is always non-negative but its empirical estimator $\hat{\mathcal R}^2_s$ defined in (12) and then is elaborated in (15) is not guaranteed to be nonnegative due to the negative term $- \frac{a\pi_n}{n_n} \sum_{i=1}^{n_n}\ell(g(x_i^n),+1)$. We explained in the paper that, as pointed out by Kiryo et al., 2017, this can lead to serious overfitting problems (we refer the readers to Figure 1 of Kiryo et al., 2017 for an illustrative experimental result). Hence, Theorem 1 is to provide methods to choose the regularization parameters such that the nonnegativity of the objective of (15) is guaranteed.
>
> -  We acknowledge the reviewer's suggestion. However, we believe that the difference between the proposed shallow and deep model is clear in the paper. For the shallow rAD model, the decision function is $g(x)=\langle w, \phi(x)\rangle$, where $\phi:\mathbb R^d \to \mathbb R^q$ is a feature map transformation, while $g(x)=\phi(x;\mathcal W)$, where $\mathcal W$ is a set of weights of a deep neural network, for the deep rAD model.
>
> - Actually, we have also tested deep model with (15) and shallow model with (14) but in our preliminary tests, they do not perform better than the proposed models in the paper. Following the reviewer's comment, we have noted this observation in the revised version.
>
> - The Equation (15) is not inherently better than Equation (14). In fact, in our initial numerical findings, we have observed that the shallow model in (18) frequently yields better results compared to the shallow model with (14), while the deep model in (19) outperforms the deep model with (15). For the shallow model, Theorem 1 (iii) provides methods to choose suitable regularization to guarantee the non-negativity of the objective of (15), but it is still open to us how to choose regularization to guarantee the non-negativity for the deep case. This fact partially explains that observation.
> If we come across the anomaly detection task, we recommend to choose the shallow model with (15) and the deep model with (14).
>
> - Following the reviewer's comment, in the revised version, we have edited the notations of Section 2 (of the old version).
>
> Reference:
>
> R. Kiryo, G. Niu, M.D du Plessis, and M. Sugiyama. Positive-unlabeled learning with non-negative risk estimator. In I. Guyon, U. Von Luxburg, S. Bengio, H. Wallach, R. Fergus, S. Vishwanathan, and R. Garnett (eds.), Advances in Neural Information Processing Systems, volume 30. Curran Associates, Inc., 2017.

---

### Decision · Action_Editor_kMU9 · 2024-02-29

**Recommendation:** Accept with minor revision

**Comment:**

The paper proposes two semi-supervised anomaly detection methods based on risk estimators - a shallow method using an unbiased risk estimator and a deep method using a biased risk estimator.

Reviewers found the paper to be well-written and easy to follow. The theoretical foundations and analysis were seen as major strengths. However, some concerns were raised about the experimental baselines being dated and the application scope being limited.

In response, the authors acknowledged some limitations and made several changes. They provided more details on the experimental setup and negative samples. The necessity of theoretical proofs was clarified. Discussion was added on handling large data and optimization challenges. Notations were improved and experiments moved to the main text.

While one reviewer still felt the baselines could be updated, others felt the paper addressed their initial feedback well. On balance, most reviewers felt the paper demonstrated novel contributions and recommended acceptance, pending minor revisions.

In summary, through discussion and revision the paper was strengthened. While some open questions remain, the novel methods, solid foundations and positive evaluation overall suggest the work represents a solid step forward for semi-supervised anomaly detection. I recommend "accept with minor revision".

To be clear, the revisions required for acceptance include the addition of recent baselines, both shallow and deep ones, for comparison. It is important to note that the request is not for state-of-the-art performance, but rather to evaluate and understand how the method performs against recent and robust baselines.

Actually I'm not convinced by the response:
> In terms of shallow anomaly detection (AD) methods, despite the introduction of OC-SVM in 2001, it remains among the top unsupervised shallow approaches. Notably, Han et al. (2022) highlighted it as one of the three representative shallow methods alongside Isolation Forest (IForest) (Liu et al., 2008) and Empirical-Cumulative-distribution-based Outlier Detection (ECOD) (Li et al., 2023). According to Table D4 and Section 4.2 of Han et al. (2022), there is no statistically significant superiority among these three methods. Given our paper’s focus on overcoming the impractical assumption of one-class classification AD methods, we opt for OC- SVM as a baseline unsupervised method in our experiments. It’s worth noting that, to the best of our knowledge, even though semi-supervised OC-SVM was introduced in 2010, it, along with PU methods, stands as a state-of-the-art choice for semi-supervised shallow AD methods, as discussed in (Ruff et al., 2021).


- Just because a method is described as "representative" does not necessarily mean it is effective or of good accuracy. Being representative could simply indicate that the method was one of the earliest or most widely referenced in its category.
- I skimmed (Ruff et al., 2021) and didn't find the statement that "semi-supervised OC-SVM ... along with PU methods, stands as a state-of-the-art choice for semi-supervised shallow AD methods." Therefore, I suggest a more thorough justification of the chosen baselines.


> Concerning deep AD methods, we acknowledge the reviewer’s suggestion to incorporate additional baseline methods in our experiments. However, as our main contribution revolves around a more in-depth exploration of risk estimation, an approach that has been relatively underexplored in the realm of AD, we *believe* that utilizing deep SAD and the deep PU learning methods as baselines is pertinent for illustrating the effectiveness of our proposed deep risk-based AD method.

I don't understand the rationale or the process behind forming this belief.

**Audience:**

Yes

**Claims And Evidence:**

Yes, mostly

---

> ### Author Response · Authors · 2024-03-18
> **Response letter**
>
> We sincerely thank the action editor for taking the time to carefully read the paper and for the additional comments.
>
> - Following the comments, in the revised version, we have added more baseline methods and improve the experiment section. Specifically, we have added recent AD methods COPOD (Li et al. 2020) and ECOD (Li et al. 2022) to the experiment for the shallow rAD model, and LOE (Qiu et al. 2022) to the experiment for the deep rAD model.
>
> - According to Section II - C3 "The Semi-Supervised Setting" of (Ruff et al., 2021), it seems that semi-supervised OC-SVM (reference [237] in the paper) and Learning from Positive and Unlabeled Examples (a.k.a PU methods) are regarded as state-of-the-art shallow anomaly detection in semi-supervised scenarios.
>
>
> Reference:
>
> Z. Li, Y. Zhao, N. Botta, C. Ionescu, and X. Hu. COPOD: copula-based outlier detection. In IEEE International Conference on Data Mining (ICDM), 2020
>
> Z. Li, Y. Zhao, X. Hu, N. Botta, C. Ionescu, and G. Chen. ECOD: Unsupervised outlier detection using empirical cumulative distribution functions. IEEE Transactions on Knowledge and Data Engineering, 2022.
>
> C. Qiu, A. Li, M. Kloft, M. Rudolph, and S. Mandt. Latent outlier exposure for anomaly detection with contaminated data. Proceedings of the 39th International Conference on Machine Learning, 2022.
>
> L. Ruff, J. Kauffmann, R. Vandermeulen, G. Montavon, W. Samek, M. Kloft, T. Dietterich, and K.-R.Müller. A unifying review of deep and shallow anomaly detection. Proceedings of the IEEE, PP:1–40, 02 2021.